# Relish plays a dynamic role in the niche to modulate *Drosophila* blood progenitor homeostasis in development and infection

**Parvathy Ramesh[1,2], Nidhi Sharma Dey[1,2†], Aditya Kanwal[1,2], Sudip Mandal[1,3], Lolitika Mandal[1,2]***

[1]Department of Biological Sciences, Indian Institute of Science Education and Research (IISER) Mohali, Knowledge City, India; [2]Developmental Genetics Laboratory, IISER Mohali, SAS Nagar, Punjab, India; [3]Molecular Cell and Developmental Biology Laboratory, IISER Mohali, SAS Nagar, Punjab, India

**\*For correspondence:**
lolitika@iisermohali.ac.in

**Present address:** [†]York Biomedical Research Institute, Hull York Medical School, University of York, York, United Kingdom

**Competing interests:** The authors declare that no competing interests exist.

**Abstract** Immune challenges demand the gearing up of basal hematopoiesis to combat infection. Little is known about how during development, this switch is achieved to take care of the insult. Here, we show that the hematopoietic niche of the larval lymph gland of *Drosophila* senses immune challenge and reacts to it quickly through the nuclear factor-κB (NF-κB), Relish, a component of the immune deficiency (Imd) pathway. During development, Relish is triggered by ecdysone signaling in the hematopoietic niche to maintain the blood progenitors. Loss of Relish causes an alteration in the cytoskeletal architecture of the niche cells in a Jun Kinase-dependent manner, resulting in the trapping of Hh implicated in progenitor maintenance. Notably, during infection, downregulation of Relish in the niche tilts the maintenance program toward precocious differentiation, thereby bolstering the cellular arm of the immune response.

## Introduction

The larval blood-forming organ, the lymph gland, is the site for definitive hematopoiesis in *Drosophila* (*Banerjee et al., 2019*; *Evans et al., 2003*; *Jung et al., 2005*; *Lanot et al., 2001*; *Mandal et al., 2004*). Interestingly, there are noticeable similarities between the molecular mechanisms that regulate the lymph gland and those essential for progenitor-based hematopoiesis in vertebrates (*Evans et al., 2003*; *Gold and Brückner, 2014*). The lymph gland is formed in embryonic stages, and through various larval stages, it grows in size. The mature third-instar larval lymph gland is a multi-lobed structure with well-characterized anterior lobe/primary lobes with three distinct zones. The heterogeneous progenitor cells (*Baldeosingh et al., 2018*; *Cho et al., 2020*) are medially located and define the medullary zone (MZ), while the differentiated hemocytes populate the peripheral zone or cortical zone of the primary lobe (*Jung et al., 2005*). The innermost core progenitors are maintained by the adjacent cardiac cells that serve as niche (*Destalminil-Letourneau et al., 2021*), while the bulk of primed progenitors are maintained by the posterior signaling center (PSC) or the niche (*Baldeosingh et al., 2018*; *Sharma et al., 2019*). Except for one study that claims otherwise (*Benmimoun et al., 2015*), several studies demonstrate that PSC/niche maintains the homeostasis of the entire organ by positively regulating the maintenance of these progenitors (*Figure 1A and B*; *Jung et al., 2005*; *Kaur et al., 2019*; *Krzemień et al., 2007*; *Mandal et al., 2007*; *Mondal et al., 2011*; *Sharma et al., 2019*). During development, this organ is the site of proliferation, maintenance, and differentiation of hemocytes. Only with the onset of pupation do the lymph glands rupture to disperse the blood cells into circulation (*Grigorian et al., 2011*).

It is fascinating to note how this reserve population within the lymph gland is prevented from precociously responding to all of the environmental challenges during normal development.

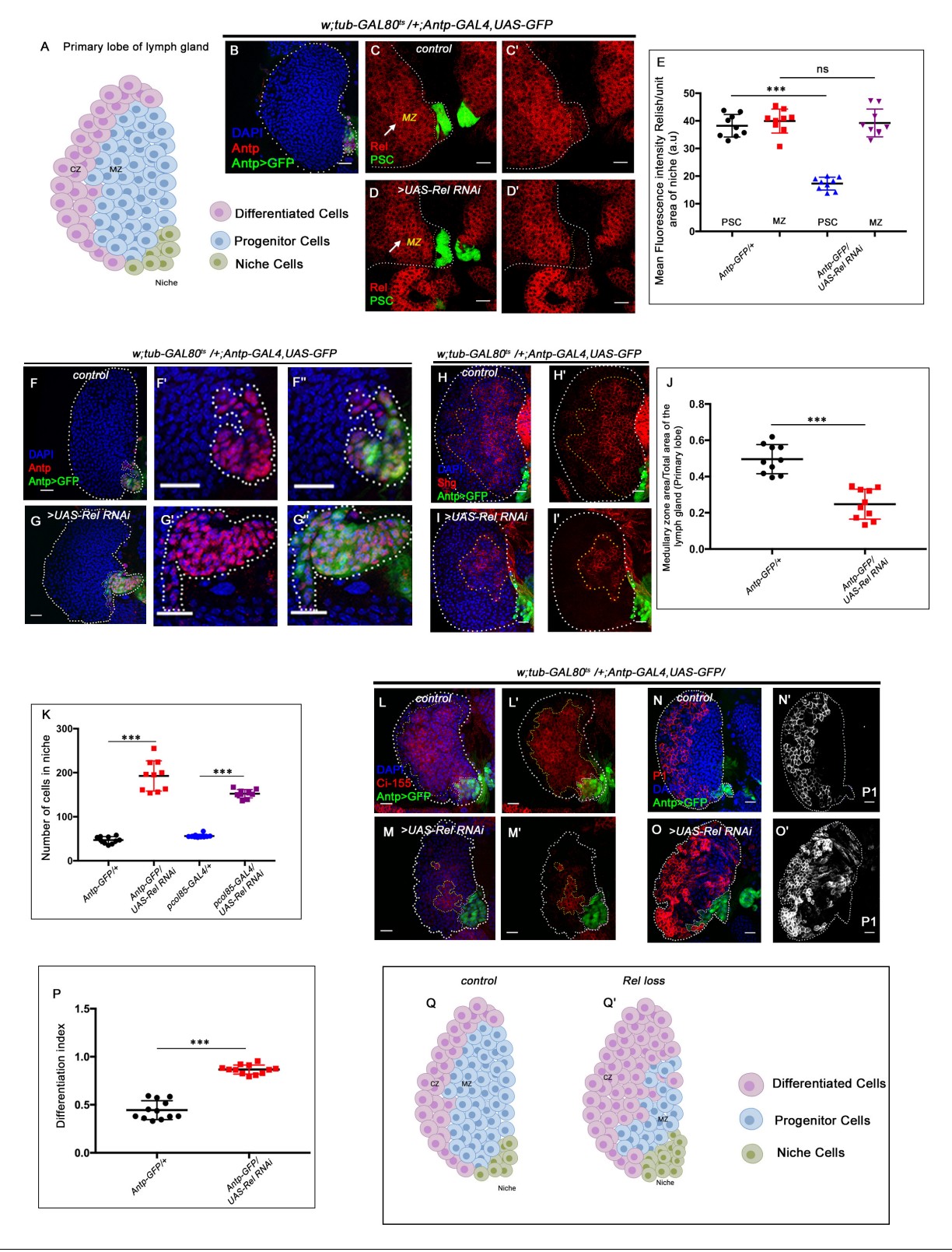

**Figure 1.** Relish expression and its function in hematopoietic niche of *Drosophila* larval lymph gland. Genotypes are mentioned in relevant panels. Scale bar: 20 μm. (**A**) Schematic representation of *Drosophila* larval lymph gland with its different cell types. (**B**) Hematopoietic niche in larval lymph gland visualized by *Antp-Gal4,UAS-GFP* and Antennapedia (Antp) antibody. (**C–D′**) Expression of Relish (antibody: red) in larval lymph gland. (**C**) Relish is expressed in the hematopoietic niche of lymph gland and in the progenitor population. (**C′**) Zoomed in view of the niche showing the expression of

*Figure 1 continued on next page*

*Figure 1 continued*

Relish in control niche. (**D–D'**) Relish expression is abrogated in the niche upon RNAi mediated downregulation. (**E**) Quantitation of Relish expression in the niche. Significant reduction in Relish expression was observed in niche (n=10, p-value=$7.4 \times 10^{-9}$, two-tailed unpaired Student's t-test), whereas progenitor-specific expression remained unchanged (n=10, p-value=0.764 , two-tailed unpaired Student's t-test). (**F–G''**) Effect of Relish loss from the niche on cell proliferation (**F–F''**), Antp expression marks the niche of wild-type lymph gland. (**G–G''**) Loss of Relish function from niche leads to increase in niche cell number. (**H–I'**) Hematopoietic progenitors of larval lymph gland (red, reported by DE-Cadherin [Shg] immunostaining). Compared to control (**H–H'**), drastic reduction in progenitor pool was observed when Relish function was attenuated from niche (**I–I'**). (**J**) Quantitation of Shg-positive progenitor population upon Relish knockdown from the niche using *Antp-GAL4* (n=10, p-value=$8.47 \times 10^{-6}$, two-tailed unpaired Student's t-test). (**K**) Quantitation of niche cell number upon Relish knockdown from the niche using *Antp-GAL4* (n=10, p-value=$1.3 \times 10^{-7}$, two-tailed unpaired Student's t-test) and *pcol85-GAL4* (n=11, p-value=$1.2 \times 10^{-12}$, two-tailed unpaired Student's t-test). (**L–M'**) Hematopoietic progenitors of larval lymph gland (red, reported by $Ci^{155}$ immunostaining) (**L–L'**). Loss of Relish from the niche resulted in reduction in $Ci^{155}$-positive progenitor pool (**M–M'**). (**N–O'**) Compared to control (**N–N'**), increase in the amount of differentiated cell population (red, P1 immunostaining) was observed upon niche-specific downregulation of Relish (**O–O'**). (**P**) Quantitative analysis of (**N–O'**) reveals significant increase in the amount of differentiated cells in comparison to control (n=10, p-value=$2.3 \times 10^{-9}$, two-tailed unpaired Student's t-test). (**Q–Q'**) Scheme based on our observation. The white dotted line mark whole of the lymph gland in all cases and niche in (**F–G''**). Yellow dotted lines mark the progenitor zone in (**H–I'**) and (**L–M'**). In all panels, age of the larvae is 96 hr AEH. The nuclei are marked with DAPI (blue). Error bar: standard deviation (SD). Individual dots represent biological replicates. Data are mean ± SD. *p<0.05, **p<0.01, and ***p<0.001.

The online version of this article includes the following source data and figure supplement(s) for figure 1:

**Source data 1.** Contains numerical data plotted in *Figure 1C–D'*, *Figure 1F–G''*, *Figure 1H–I'*, *Figure 1N–O'* and *Figure 1—figure supplement 1C–D'*, *Figure 1—figure supplement 1F–G''* and *Figure 1—figure supplement 1K–L'*.

**Figure supplement 1.** Relish negatively regulate niche cell proliferation.

Interestingly, during infection, the lymph gland releases the differentiated hemocytes into circulation in larval stages (*Khadilkar et al., 2017*; *Lanot et al., 2001*; *Louradour et al., 2017*; *Sorrentino et al., 2002*).

The three *Drosophila* NF-κB factors – Dorsal, Dorsal-related immunity factor (DIF), and Relish – regulate the insect humoral immunity pathway that gets activated during infection (*Govind, 1999*; *Hetru and Hoffmann, 2009*; *Louradour et al., 2017*). *Drosophila* NF-κB signaling pathways show conspicuous similarity with vertebrates. The NF-κB family consists of five members – RelA (p65), RelB, c-Rel, p50/p105, and p52/p100 (*Ganesan et al., 2010*). In vertebrates, these factors are critical for producing cytokines, regulating cell death, and controlling cell cycle progression (*Gilmore, 2006*). In *Drosophila*, Dorsal and Dif activation happens during embryogenesis as well as during gram-positive bacterial and fungal infections. In both cases, it is triggered by the activation of the Toll pathway by cleaved cytokine Spatzle (*Valanne et al., 2011*).

On the other hand, gram-negative bacterial infections activate the Imd pathway. The diaminopimelic acid (DAP)-type peptidoglycan from the cell wall of the bacteria directly binds to the peptidoglycan recognition protein-LC (*Choe et al., 2002*; *Gottar et al., 2002*; *Kaneko et al., 2006*; *Rämet et al., 2002*) or peptidoglycan recognition protein-LE (PGRP-LC or PGRP-LE). This binding initiates a signaling cascade that elicits the cleavage, activation, and nuclear translocation of Relish with the subsequent transcription of antimicrobial peptide genes (*Choe et al., 2002*; *Hedengren et al., 1999*).

IMD pathway has been studied intensively in the context of immunity and inflammation, but far less is understood about the developmental function of this pathway. Accumulating evidence from studies, however, suggests that the IMD pathway may also have distinct roles in development. For example, in *Drosophila*, Relish and its target genes are activated during neurodegeneration and overexpression of Relish during development causes apoptosis in wing disc cells, neurons, photoreceptors (*Cao et al., 2013*; *Chinchore et al., 2012*; *Katzenberger et al., 2013*; *Tavignot et al., 2017*) and autophagy in salivary gland cell (*Nandy et al., 2018*). These studies point out to diverse developmental requirements of Relish beyond immunity in *Drosophila*. Since IMD is an evolutionarily conserved signaling cascade, *Drosophila*, therefore, turns out to be a great model to explore the diverse function of the components of this pathway.

Expression of Relish in the hematopoietic niche of the lymph gland during non-infectious conditions prompted us to investigate its role in developmental hematopoiesis. We found that Relish acts as an inhibitor of c-Jun Kinase Signaling (*JNK*) in the hematopoietic niche. During infection, Relish inhibits JNK signaling through *tak1* in *Drosophila* (*Park et al., 2004*). Interestingly, we found similar crosstalk being adopted during development in the hematopoietic niche. Activation of JNK signaling

in *Drosophila* is associated with alteration of the cytoskeletal architecture of cells during various developmental scenarios, including cell migration, dorsal closure, etc (*Homsy et al., 2006*; *Jacinto et al., 2000*; *Kaltschmidt et al., 2002*; *Kockel et al., 2001*; *Rudrapatna et al., 2014*). We found that upon Relish loss, JNK activation causes upregulation of actin remodelers, Enabled and Singed in the niche. The actin cytoskeletal remodeling, in turn, affects the formation of cytoneme-like filopodial projections leading to precocious differentiation at the expense of progenitors. These filopodial projections are proposed to facilitate the transporting of Hh from the niche to the adjoining progenitors (*Mandal et al., 2007*). We further show that perturbation in filopodial extensions via downregulation of Diaphanous affects Hh delivery and disrupts the communication between niche and progenitors. The hematopoietic niche maintains the delicate balance between the number of progenitors and differentiated cells of the lymph gland (*Baldeosingh et al., 2018*; *Krzemień et al., 2007*; *Mandal et al., 2007*; *Sharma et al., 2019*). During development, this organ accumulates hemocytes for post-larval requirements. However, during wasp infestation, this organ precociously releases the content into circulation (*Lanot et al., 2001*) due to the activation of the Toll pathway in the PSC/hematopoietic niche (*Louradour et al., 2017*). Therefore, a switch is essential to enable the transition from basal hematopoiesis toward the emergency mode to enable the organism to combat infection. The pathway identified in this study, critical for niche maintenance and developmental hematopoiesis, is also exploited during the immune challenge. The circuit engaged in niche maintenance and, therefore, crucial for developmental hematopoiesis gets disrupted during bacterial infection. We found that Relish in the niche serves as a joystick to achieve control between developmental and immune response.

Previous studies have demonstrated that Relish needs to be activated in the fat body to mount an immune response (*Cha et al., 2003*; *Charroux and Royet, 2010*). We show that to reinforce the cellular arm of the innate immune response, Relish needs to be downregulated in the niche during infection. Though the candidate that breaks the maintenance circuit remains to be identified, nonetheless, our study illustrates that the hematopoietic niche can sense the physiological state of an animal to facilitate a transition from normal to emergency hematopoiesis.

## Results

### The hematopoietic niche requires Relish during development

*Drosophila* NF-κB-like factor, Relish, has been studied extensively as a major contributor of humoral immune defense mechanism against gram-negative bacterial infections (*Buchon et al., 2014*; *Ferrandon et al., 2007*; *Ganesan et al., 2010*; *Gottar et al., 2002*; *Kleino and Silverman, 2014*). During larval development, Relish expresses in the hematopoietic niche (marked by *Antp-GAL4>UAS-GFP*, a validated reporter for niche cells; *Figure 1C–C'*). In addition to the niche, the hemocyte progenitor cells (MZ) also express Relish (arrow, MZ, *Figure 1C*). The niche-specific expression was further validated by the downregulation of Relish using *UAS-Relish RNAi* within the niche that resulted in complete loss of Relish protein therein (*Figure 1D–D'*). As evident from the quantitative analysis (*Figure 1E*) of the above data, the expression of Rel in the niche was drastically affected, while that of the MZ is comparable to the control. Whether this transcription factor executes any role in developmental hematopoiesis, beyond its known role in immune response, inspired us to carry out in vivo genetic analysis using *Drosophila* larval lymph gland.

We employed the TARGET system (*McGuire et al., 2004*) to investigate the role of Relish, if any, in the hematopoietic niche. Compared to the control, wherein the number of cells in the hematopoietic niche ranges from 40 to 45 (*Figure 1F–F'' and K*), a niche-specific downregulation of Relish results in a fourfold increase in the cell number (*Figure 1G–G'' and K*). A similar increase is evidenced upon downregulation of Relish by another independent niche-specific driver, *collier-GAL4* (*Krzemień et al., 2007*; *Figure 1—figure supplement 1A–B'* and *Figure 1K*). To further validate the phenotype, the lymph gland from the classical loss of function of Relish (*Rel^{E20}*) was analyzed. Interestingly, compared to control, *Rel^{E20}* niches exhibit a twofold increase in cell number (*Figure 1—figure supplement 1C-D' and E'*). Likewise, overexpression of Relish specifically, in the niche, causes a decline in the niche cell number (*Figure 1—figure supplement 1F-G'' and H*).

To investigate whether the hyperproliferative niche is still capable of performing its function of progenitor maintenance (*Mandal et al., 2007*), we assayed the status of the progenitors.

Interestingly, compared to the control, the loss of Relish from the niche results in a drastic reduction in the number of the progenitor cells (visualized by DE-Cadherin: Shg *Jung et al., 2005*; *Sharma et al., 2019*; *Figure 1H–I' and J*) and Cubitus interruptus: Ci[155] (*Figure 1L–M'*) with a concomitant increment in the number of differentiated hemocytes (visualized by plasmatocyte marker by P1, Nimrod; *Figure 1N–O'*; *Asha et al., 2003*; *Jung et al., 2005*; *Kurucz et al., 2007*). Quantitation of differentiation index in the genotype described above reveals a twofold increase in plasmatocyte number (*Figure 1P*). Moreover, in these lymph glands, the differentiated cells, instead of being spatially restricted in the CZ, are dispersed throughout (*Figure 1N–O'*).

Although the differentiation index increases, there was no induction of lamellocytes (visualized by lamellocyte marker β-PS: myospheriod; *Stofanko et al., 2008*; *Figure 1—figure supplement 1I–J'*). The crystal cell numbers also remain unaltered (*Figure 1—figure supplement 1K-L' and M*), suggesting a tilt toward plasmatocyte fate upon Relish loss from the niche.

These results collectively indicate that Relish plays a critical role in determining the number of niche cells in the developing lymph gland (*Figure 1Q–Q'*).

## Relish loss from the hematopoietic niche induces proliferation

Our expression analysis throughout development reveals that around 45–48 hr AEH (after egg hatching), Relish can be detected in the niche as well as in the progenitors (*Figure 2—figure supplement 1A-E'*). The co-localization of Rel with validated markers of progenitors like TepIV (*Dey et al., 2016*; *Irving et al., 2005*; *Kroeger et al., 2012*; *Shim et al., 2013*) and Ance (*Benmimoun et al., 2012*; *Sharma et al., 2019*) further endorsed Rel's progenitor-specific expression (*Figure 2—figure supplement 1F-F''*). On the other hand, co-labeling with Pxn-YFP, a differentiated cell marker (*Nelson et al., 1994*), reveals that Rel is downregulated from CZ (*Figure 2—figure supplement 1G-G'*). Therefore, we traced back to post second-instar stages to get a better insight into the phenotype caused by Relish loss from the niche. At 54–64 hr AEH, compared to wild type (*Figure 2A–A'', C–C'', and I*), downregulation of Rel by *Antp-Gal4* results in an increase in EdU incorporation in the niche (*Figure 2B–B'', D–D'', and I*). In context to the niche, a definite proliferation pattern is observable during development. Compared to the rest of the lymph gland, niche cell proliferation decreases by 86 hr AEH (*Figure 2E–E'' and I*). Beyond this time point, EdU incorporation rarely occurs in the niche (*Figure 2G–G'' and I*). In sharp contrast to this, upon niche-specific downregulation of Relish, there is a failure in attaining the steady-state proliferative pattern by 86 hr AEH (*Figure 2F–F'' and I*). Quite strikingly, EdU incorporation continues even at 96 hr when the control niche cells have stopped proliferating (compare *Figure 2G–G''* with H–H'' and *Figure 2I*). These proliferating niche cells are indeed mitotically active is evident by the increase in phospho histone H3 (PH3) incorporation compared to the control (*Figure 2J–K'' and L*). In addition to these snapshot techniques, in vivo cell proliferation assay of the niche was done employing the FUCCI system (fluorescent ubiquitination-based cell cycle indicator) (*Zielke and Edgar, 2015*). Fly-FUCCI relies on fluorochrome-tagged probes where the first one is a GFP fused to E2F protein, which is degraded at the S phase by Cdt2 (thus GFP marks cells in G2, M, and G1 phases). The second probe is an mRFP tagged to the CycB protein, which undergoes anaphase promoting complex/cyclosome-mediated degradation during mid-mitosis (thereby marking cells in S, G2, and M phases). While in control by 96 hr AEH, niche cells are mostly in G2-M (yellow), and in G1 state (green), in loss of Relish, abundance in S phase can be seen at the expense of G1 (*Figure 2—figure supplement 1H-I'''' and J*).

Put together, these results implicate that Relish functions as the negative regulator of niche proliferation in the developing lymph gland.

## Absence of Relish in the niche stimulates proliferation via upregulation of Wingless signaling

Previous studies have shown that the Wingless (Wg) pathway positively regulates niche cell number in addition to its role in the maintenance of the prohemocyte population in the MZ (*Sinenko et al., 2009*). Upon perturbation of Relish function, a drastic increase in the level of Wingless is evident (arrow, *Figure 3B–B''*) in the niche compared to the control (arrow, *Figure 3A–A''*). Quantitative analysis reveals a 1.6-fold increase in the fluorescence intensity of Wg per unit area in the niche where Rel function is attenuated compared to that of the control (*Figure 3C*). Tweaking of Wg in the background of Rel loss from the niche by RNAi constructs led to a decline in niche cell number

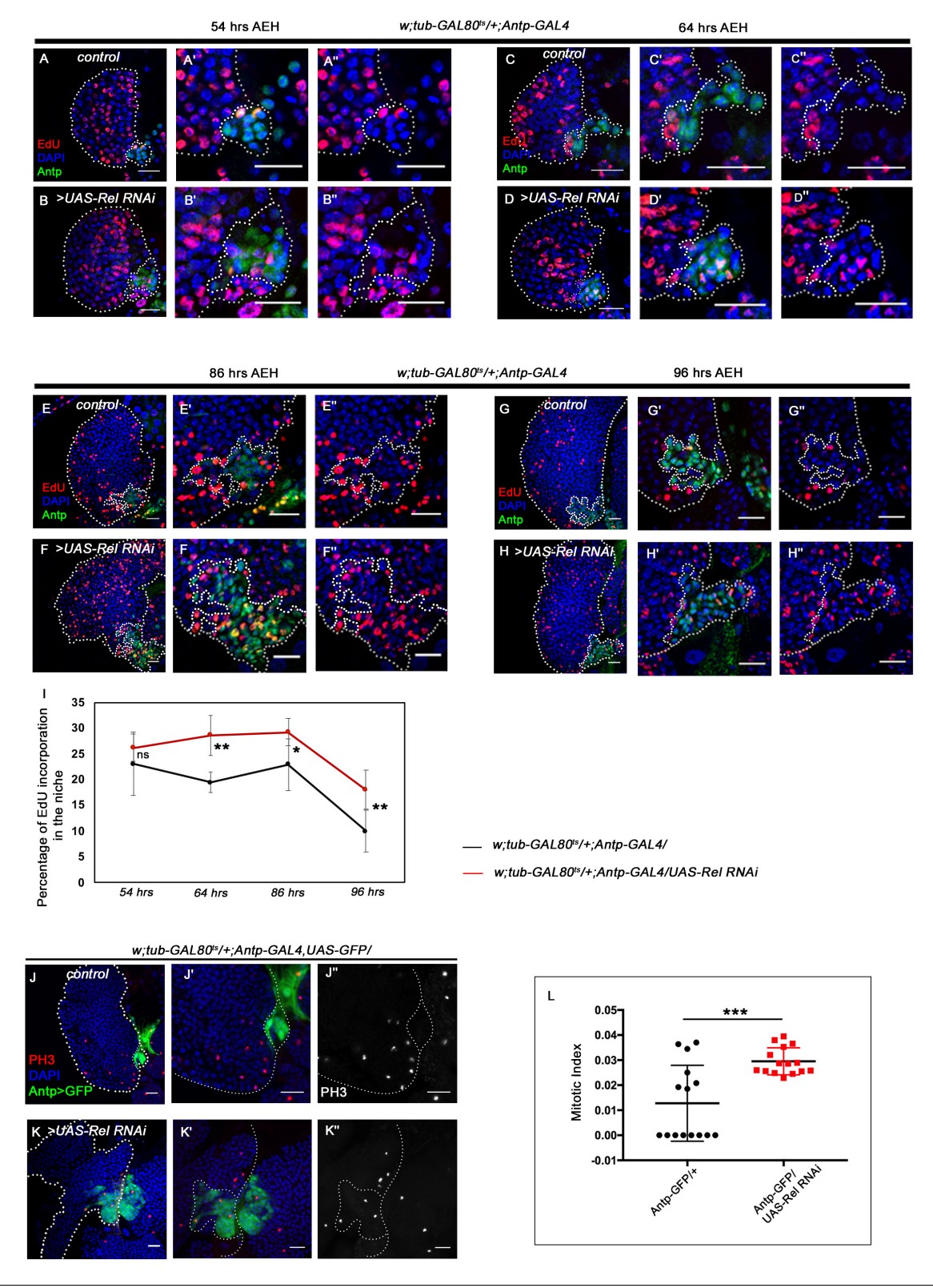

**Figure 2.** Loss of Relish from the niche causes niche cell hyperplasia. Genotypes are mentioned in relevant panels. Scale bar: 20 μm. Niche is visualized by Antp antibody expression. (**A–H''**) EdU or 5-ethynyl-2'-deoxyuridine marks the cells in S-phase of the cell cycle. EdU profiling at 54 hr AEH (**A–B''**), 64 hr AEH (**C–D''**), 86 hr AEH (**E–F''**), and 96 hr AEH (**G–H''**) displayed EdU incorporation in the niche (green) in control and upon Relish downregulation. Control niches showed scanty EdU incorporation beyond 84 hr (**E–E''** and **G–G''**), whereas loss of Relish induced niche cells to proliferate more (**F–F''**

*Figure 2 continued on next page*

*Figure 2 continued*

and H–H''). (I) Graph representing percentage of EdU incorporation in the niche during the course of development in control (black line) and Relish loss (red line). Significant increase in the niche cell number is observed with development in Relish loss scenario. (54 hr, n=6, p-value=0.294), (64 hr, n=6, p-value=1.3 × 10$^{-3}$), (86 hr, n=6, p-value=2.9 × 10$^{-2}$), (96 hr, n=6, p-value=5.9 × 10$^{-3}$); two-tailed unpaired Student's t-test. (J–K'') Significant increase in the number of mitotic cells (phospho-histone 3 [PH3], red) was observed upon Relish loss from the niche (K–K'') compared to the control (J–J''). (L) Quantitation of the mitotic index of wild-type and Relish loss niche (n=15, p-value=8.1 × 10$^{-4}$; two-tailed unpaired Student's t-test). The white dotted line marks whole of the lymph gland and the niches. In all panels, age of the larvae is 96 hr AEH, unless otherwise mentioned. The nuclei are marked with DAPI (blue). Individual dots represent biological replicates. Error bar: standard deviation (SD). Data are mean ± SD. *p<0.05, **p<0.01, and ***p<0.001.

The online version of this article includes the following source data and figure supplement(s) for figure 2:

**Source data 1.** Contains numerical data plotted in *Figure 2A–B''*, *Figure 2C–D''*, *Figure 2E–F''*, *Figure 2G–H''*, *Figure 2J–K''* and *Figure 2—figure supplement 1H–I''''*.

**Figure supplement 1.** Relish expression starts beyond the second-instar stage in the hematopoietic niche. The genotypes are mentioned in relevant panels.

compared to Rel loss from the niche (compare *Figure 3G–G'* with *Figure 3E–E' and H*), restores the hyperproliferative niche to a cell number comparable to the control (*Figure 3D–D' and H*).

Interestingly, although the niche cell number was restored in the above genotype, the defects in the maintenance of progenitors (*Figure 3J–M and N*) and differentiation (*Figure 3—figure supplement 1A–D and E*) observed upon Relish loss from the niche were still evident.

Similarly, reducing Wg by using a temperature-sensitive mutant allele *wg$^{ts}$ Bejsovec and Martinez Arias, 1991* following the scheme provided in *Figure 3—figure supplement 1F*, gave similar restoration of the hyperproliferative phenotype (*Figure 3—figure supplement 1G–K*). In this case also, there was a failure in rescuing the defects in progenitor maintenance (*Figure 3—figure supplement 1L-O and P*) as well as differentiation (*Figure 3—figure supplement 1Q–T and U*).

This set of experiments led us to infer that the upregulated Wg in Relish loss was responsible only for controlling the niche cell number.

## In the absence of Relish, altered cytoskeletal architecture of the niche traps Hh

Various studies have established PSC as the niche for hematopoietic progenitors and have shown that it employs a morphogen Hedgehog for its maintenance. It has also been shown that niche expansion correlates to expansion in the progenitor population (*Baldeosingh et al., 2018*; *Benmimoun et al., 2012*; *Mandal et al., 2007*; *Pennetier et al., 2012*; *Tokusumi et al., 2011*). However, in contrast to the above studies, despite a threefold increment in niche cell number upon Relish downregulation, we observed a significant reduction in the progenitor pool (*Figure 1L–M'*). Moreover, restoration in the number of niche cells by modulating Wg levels in Relish knockdown condition failed to restore the differentiation defects observed upon downregulating Relish from the niche (*Figure 3J–M and N*, *Figure 3—figure supplement 1A–D and E*, *Figure 3—figure supplement 1L-O and P*, *Figure 3—figure supplement 1Q–T and U*). To understand this result, we assayed Hedgehog levels in the niche by using an antibody against Hh protein (*Forbes et al., 1993*). Interestingly, compared to that of the control, there is a substantial increase in Hh protein in the niche where the Relish function is abrogated (*Figure 4A–B''*). Quantitative analysis reveals an almost twofold increase in the level of Hh protein in the experimental niche (*Figure 4C*).

However, despite having a higher amount of Hh in the niche upon Relish downregulation, there was a decline in the amount of extracellular Hh (Hh$^{Ext}$) in the prohemocytes compared to control (*Figure 4D–E'' and F*). This result is in sync with the observation that Rel loss from the niche leads to the reduction in the levels of Ci$^{155}$ in the progenitors (*Figure 1L–M'*), suggesting that Hh produced by the niche is not sensed by the progenitors resulting in their precocious differentiation.

The alteration in extracellular Hh and decline in Ci$^{155}$ level in the progenitors prompted us to speculate that loss of Relish from niche might have interfered with Hh delivery to the progenitor cells. Several reports in diverse tissues across model organisms have demonstrated filopodia mediated Hh delivery (*Bischoff et al., 2013*; *González-Méndez et al., 2019*). Although the filopodial extension has been documented in the case hematopoietic niche (*Krzemień et al., 2007*; *Mandal et al., 2007*), its role in Hh delivery is yet to be demonstrated. To check this possibility, we

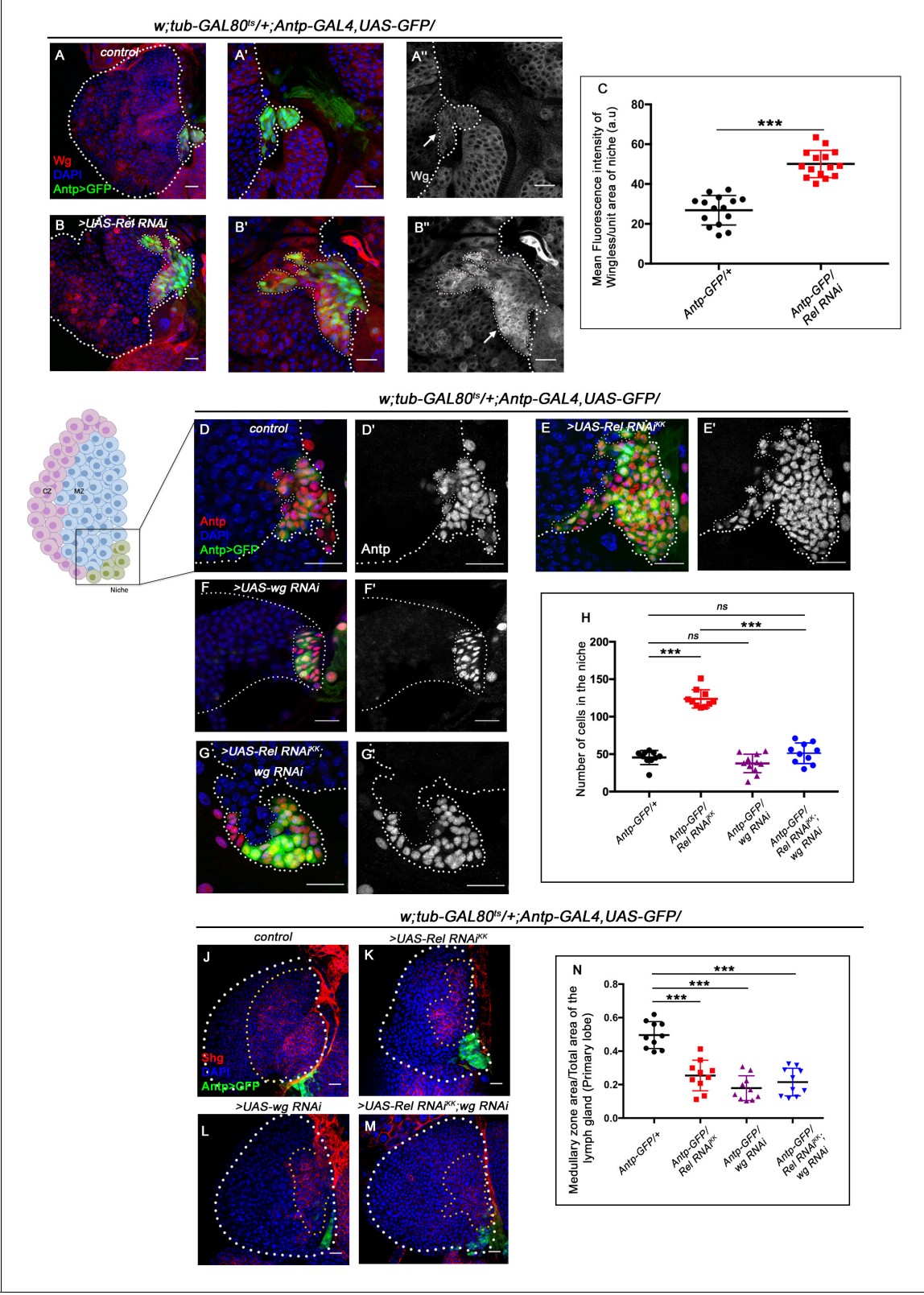

**Figure 3.** Upregulated Wingless signaling leads to increase in niche cell number. The genotypes are mentioned in relevant panels. Scale bar: 20 μm. (A–B′′) Expression of Wingless (antibody) in the lymph gland. The hematopoietic niche is visualized by *Antp-GAL4>UAS-GFP*. (A′–A′′) and (B′–B′′) are higher magnifications of (A) and (B), respectively. In comparison to the wild-type niche (A–A′′), Wingless protein levels were substantially high in Relish loss of function (B–B′′). (C) Statistical analysis reveals elevated wingless expression upon Relish knockdown in niche (n=15; p-value=5.8 × 10$^{-9}$, two-

*Figure 3 continued on next page*

*Figure 3 continued*

tailed unpaired Student's t-test.) (**D–G'**) The increased niche number observed upon Relish loss (**E–E'**) is rescued upon reducing Wingless level by the *wg RNAi* (**F–F'**) in Relish loss genetic background (**G–G'**). The rescued niche cell number is comparable to control (**D–D'**). (**H**) Statistical analysis of the data in (**D–G'**) (n=10, p-value=$1.1 \times 10^{-11}$ for control versus *Rel RNAi*[KK], p-value=$3.15 \times 10^{-10}$ for *Rel RNAi*[KK] versus *Rel RNAi*[KK]; *wg RNAi*, n=10, p-value=0.10 for control versus *wg RNAi*, n=10, p-value=0.29 for control versus *Rel RNAi*[KK]; *wg RNAi*; two-tailed unpaired Student's t-test). (**J–M**) Hematopoietic progenitors of larval lymph gland (red, reported by DE-Cadherin [Shg] immunostaining). Knocking down wingless function from the niche resulted in loss of Shg-positive progenitors (**L**). Downregulating wingless using *wg RNAi* in Relish loss genetic background was unable to restore the reduction in prohemocyte pool (**M**) observed in Relish loss (**K**) scenario in comparison to control (**J**). (**N**) Statistical analysis of the data in (**J–M**) (n=10, p-value=$6.74 \times 10^{-6}$ for control versus *Rel RNAi*[KK], p-value=$4.03 \times 10^{-7}$ for control versus *wg RNAi*; *Rel RNAi*[KK], p-value=$3.42 \times 10^{-8}$ for control versus *wg RNAi*; two-tailed unpaired Student's t-test). The white dotted line marks whole of the lymph gland and the niches in (**A–G'**) Yellow dotted lines mark the progenitor zone in (**J–M**). In all panels, age of the larvae is 96 hr AEH. The nuclei are marked with DAPI (blue). Individual dots represent biological replicates. Error bar: standard deviation (SD). Data are mean ± SD. *p<0.05, **p<0.01, and ***p<0.001.

The online version of this article includes the following source data and figure supplement(s) for figure 3:

**Source data 1.** Contains numerical data plotted in *Figure 3A–B''*, *Figure 3D–G'*, *Figure 3J–M* and *Figure 3—figure supplement 1A–D*, *Figure 3—figure supplement 1G–J*, *Figure 3—figure supplement 1L–O* and *Figure 3—figure supplement 1Q–T*.

**Figure supplement 1.** Downregulating wingless in Relish loss condition rescues niche cell proliferation, but not differentiation.

assayed the status of these actin-based cellular extensions emanating from the niche cells in freshly dissected unfixed tissue of control as well experimental. For this purpose, *UAS-GMA* (also known as *UAS-moesin-GFP*) that marks F-actin (*Kiehart et al., 2000*) was expressed in a niche-specific manner. Multiple cellular processes with variable length are detectable in control, while upon Relish knockdown, filopodial extensions are highly compromised (arrowheads, *Figure 4G–I'*). Quantitative analyses of the data reveal that both length (*Figure 4J*) and number (*Figure 4K*) are altered upon Rel loss from the niche. Intrigued with this finding, we independently downregulated Diaphanous (*dia*), an actin polymerase known to be important in filopodial formation, elongation and maintenance (*Homem and Peifer, 2009*; *Nowotarski et al., 2014*), from the niche. As expected, compared to control niches, *dia* loss resulted in compromised filopodial length and number (*Figure 4—figure supplement 1A–B and C-D*). Quite similar to *Rel* loss from the niche, these defects in filopodial in turn affected Hh delivery from the niche (*Figure 4—figure supplement 1E–F' and G*). As a consequence, there was a decline in the number of progenitors (*Figure 4—figure supplement 1H–I and L*) and a concomitant increase in the differentiated cells (*Figure 4—figure supplement 1J–K and M*) compared to control.

Additionally, compared to the control, F-actin (visualized by rhodamine-phalloidin) expression is significantly increased in the cell cortex upon Relish loss from the niche (*Figure 4—figure supplement 2A–B'' and C*). This accumulation of cortical F-actin intrigued us to further probe into F-actin associated proteins' status, Singed and Enabled in the niche cells upon loss of Relish. While Singed is the *Drosophila* homolog of Fascin and is involved in cross-linking actin filaments and actin-bundling (*Cant et al., 1994*; *Tilney et al., 2000*), Enabled is a cytoskeletal adaptor protein involved in actin polymerization (*Gates et al., 2007*; *Lin et al., 2009*). In comparison to the control, where there is a basal level of Singed or lack of Ena expression in the niche, a significant increase in the level of both of these actin-associated proteins occurs upon downregulation of Relish function (Singed: *Figure 4—figure supplement 2D–E'' and F* and Ena: *Figure 4—figure supplement 2G–H'' and I*).

Interestingly, co-expressing the RNAi construct of Ena and Rel in the niche partially rescued the defects in progenitor maintenance (*Figure 4—figure supplement 3A–C and D*) and differentiation (*Figure 4—figure supplement 3E–G and H*), which is otherwise seen upon Rel loss. This rescue in the phenotype can be attributed to the resurrection of the transport defects of Hh seen upon Rel loss from the niche (*Figure 4—figure supplement 3I–K' and L-N*).

These results demonstrate that loss of Relish from the niche induces cytoskeletal rearrangement, which disrupts the proper delivery of Hedgehog to the adjoining progenitors. These results further emphasize how aberrant cytoskeleton architecture might interfere with niche functionality by trapping Hh.

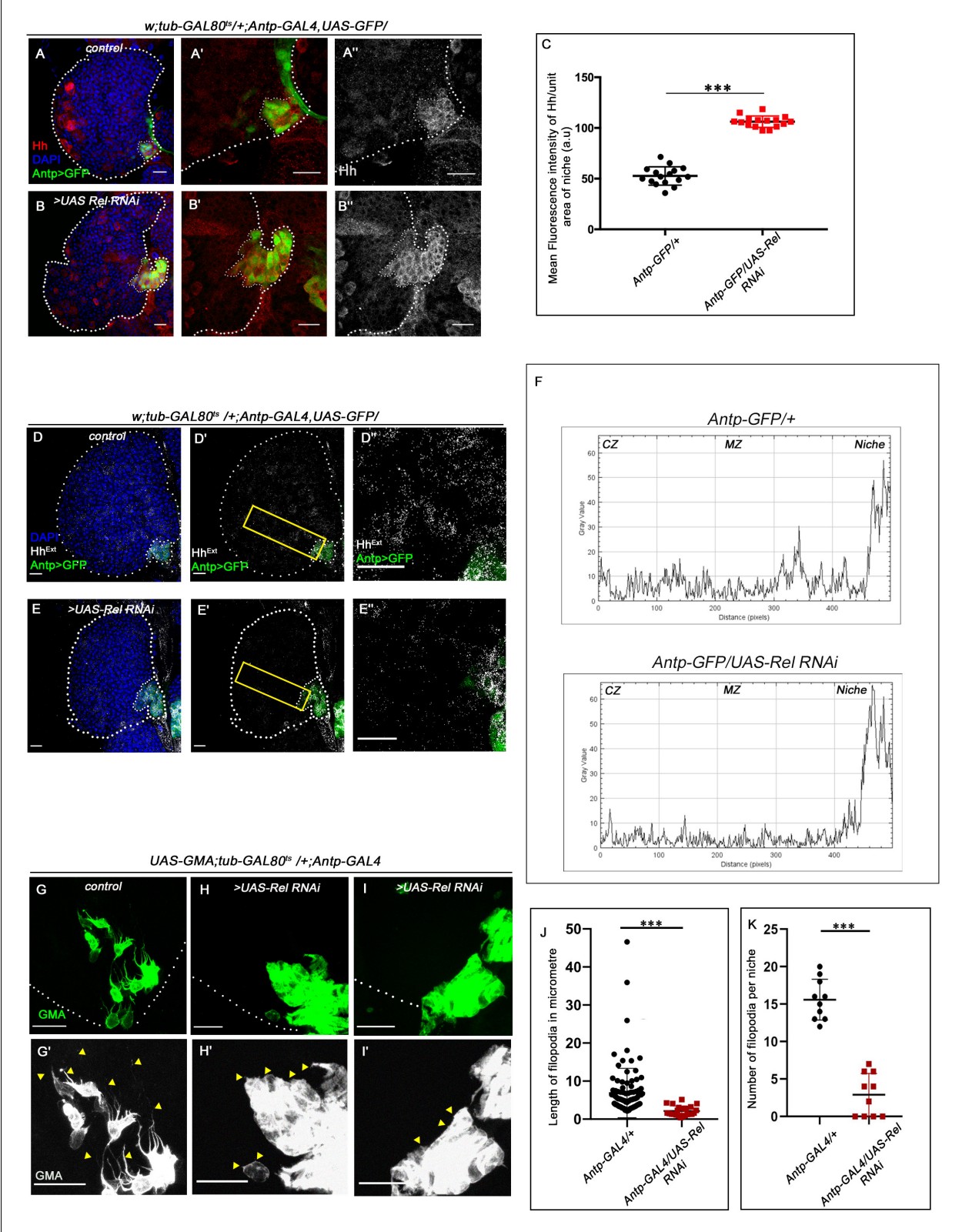

**Figure 4.** Hedgehog release from the niche is affected in Relish loss of function. The genotypes are mentioned in relevant panels. Scale bar: 20 μm. (A–B'') Hedgehog (Hh) antibody staining in the lymph gland shows Hh enrichment in the niche. The hematopoietic niche in Relish loss of function (B–B'') exhibits higher level of Hh in comparison to the control (A–A''). (C) Statistical analysis of fluorescence intensity revealed more than 2.5-fold increase in Hh levels compared to control (n=15, p-value=$2.5 \times 10^{-17}$, two-tailed Students t-test). (D–E'') Progenitors in Relish loss of function exhibits lower level

*Figure 4 continued on next page*

*Figure 4 continued*

of Extracellular Hh (Hh$^{Extra}$) (E–E'') in comparison to those of control (D–D''). (E'' and D'') are zoomed in view of niche and the neighboring progenitor cells of (E' and D'), respectively. The yellow box denotes the area quantified in (F). (F) The intensity profile of Hh$^{Extra}$ in progenitors (along the rectangle drawn from PSC to cortical zone housing differentiated cells in D' and E') reflects a stark decline in the level of Hh$^{Extra}$ in Relish loss scenario compared to control. (G–I') Cellular filopodia emanating from the niche cells were stabilized by using untagged phalloidin. The filopodia in Relish loss of function niches were found to be smaller in length and fewer in number (H–H', I–I') as compared to control (G–G'). (J–K) Significant reduction in filopodial length (J, n=10, p-value=$6.64 \times 10^{-9}$, two-tailed Student's t-test) and number (K, n=6, p-value=$9.19 \times 10^{-10}$, two-tailed Student's t-test) were observed in Relish loss scenario compared to control. The white dotted line marks whole of the lymph gland and niches in A–B'', D-E'. In all panels, age of the larvae is 96 hr AEH. The nuclei are marked with DAPI (blue). Individual dots represent biological replicates. Error bar: standard deviation (SD). Data are mean ± SD. *p<0.05, **p<0.01, and ***p<0.001.

The online version of this article includes the following source data and figure supplement(s) for figure 4:

**Source data 1.** Contains numerical data plotted in *Figure 4A–B''*, *Figure 4D–E''*, *Figure 4G–I'*, *Figure 4—figure supplement 1A–B'*, *Figure 4—figure supplement 1E–F*, *Figure 4—figure supplement 1H–I*, *Figure 4—figure supplement 1J–K*, *Figure 4—figure supplement 2A–B''*, *Figure 4—figure supplement 2D–E''*, *Figure 4—figure supplement 2G–H''*, *Figure 4—figure supplement 3A–C*, *Figure 4—figure supplement 3E–G* and *Figure 4—figure supplement 3I–K'*.

**Figure supplement 1.** Loss of Diaphanous from the niche resulted in defect in filopodial formation and enhanced differentiation.

**Figure supplement 2.** Loss of Relish from the niche resulted in upregulation of actin remodelers.

**Figure supplement 3.** Downregulation of Ena in Rel loss genetic condition partially rescues the differentiation and Hh$^{Extra}$ dispersal defects.

## Ectopic JNK activation leads to precocious differentiation in relish loss from the niche

Next, we investigated how Relish loss causes alterations in cytoskeletal architecture within the niche. Studies across the taxa have shown mitogen-activated protein kinases (MAPKs) as a major regulator of cellular cytoskeleton dynamics (*Densham et al., 2009*; *Pichon et al., 2004*; *Reszka et al., 1995*; *Šamaj et al., 2004*). The c-Jun-NH2-terminal kinase (JNK) or so-called stress-activated protein kinases, which belong to the MAPK superfamily, are one such key modulator of actin dynamics in a cell. Whether the cytoskeletal remodeling of the niche in the absence of Relish is an outcome of JNK activation was next explored. Compared to the control where there is a negligible level of activation of JNK signaling in the niche, visualized by TRE-GFP: a transcriptional reporter of JNK (*Chatterjee and Bohmann, 2012*), a robust increase in the expression occurs in the niche where the function of Relish is abrogated (*Figure 5A–B' and C*). This result implicates that during development, Relish inhibits JNK activation in the hematopoietic niche.

Interestingly, activation of JNK alone (expression of Hep$^{act}$) in the niche can recapitulate the phenotypes associated with Relish loss to a large extent, for example, hyperproliferative niche (visualized by Antp, *Figure 5—figure supplement 1A–B'*), ectopic differentiation (visualized by Nimrod P1, *Figure 5—figure supplement 1D–E' and F*), and upregulated cytoskeletal elements (visualized by Enabled, *Figure 5—figure supplement 1G–H'' and I*). Moreover, downregulating wg function in the same genetic background restores the cell number within the niche. These results further validate the epistatic relation of JNK and Wg in context to the hematopoietic niche (*Figure 5—figure supplement 1J–M and N*).

To further understand the relationship of Relish-JNK in the context of niche cell proliferation and functionality, a double knockdown of both JNK and Relish from the niche was analyzed. The concurrent loss of JNK and Relish rescues the increase in niche cell proliferation, seen upon Relish loss (*Figure 5D–G' and H*). Moreover, downregulating JNK in conjunction with Relish loss from the niche restores the abrogated filopodial extension (*Figure 5I–L*). The quantitative analyses further reveal the restoration of filopodial length (*Figure 5M*) and number (*Figure 5N*) in the above genotype. The rescue, in turn, restored the progenitor pool (*Figure 5O–R and S*) and the differentiation defect (*Figure 5—figure supplement 2A–D and E*) noted in the lymph gland upon Relish loss from the niche. The rescue in ectopic differentiation coupled with the resurrection of the filopodial extension suggests a re-establishment of the communication process between the niche and the progenitors.

To have a functional insight into this result, we checked the extracellular Hh level (Hh$^{Ext}$) in the same genetic background. We found that in the double knockdown of JNK and Relish, the level of Hh$^{Ext}$ present in the progenitors is similar to that of the control (*Figure 5—figure supplement 2F–H' and I-K*). Therefore, the downregulation of the elevated JNK in Relish loss restores niche cell

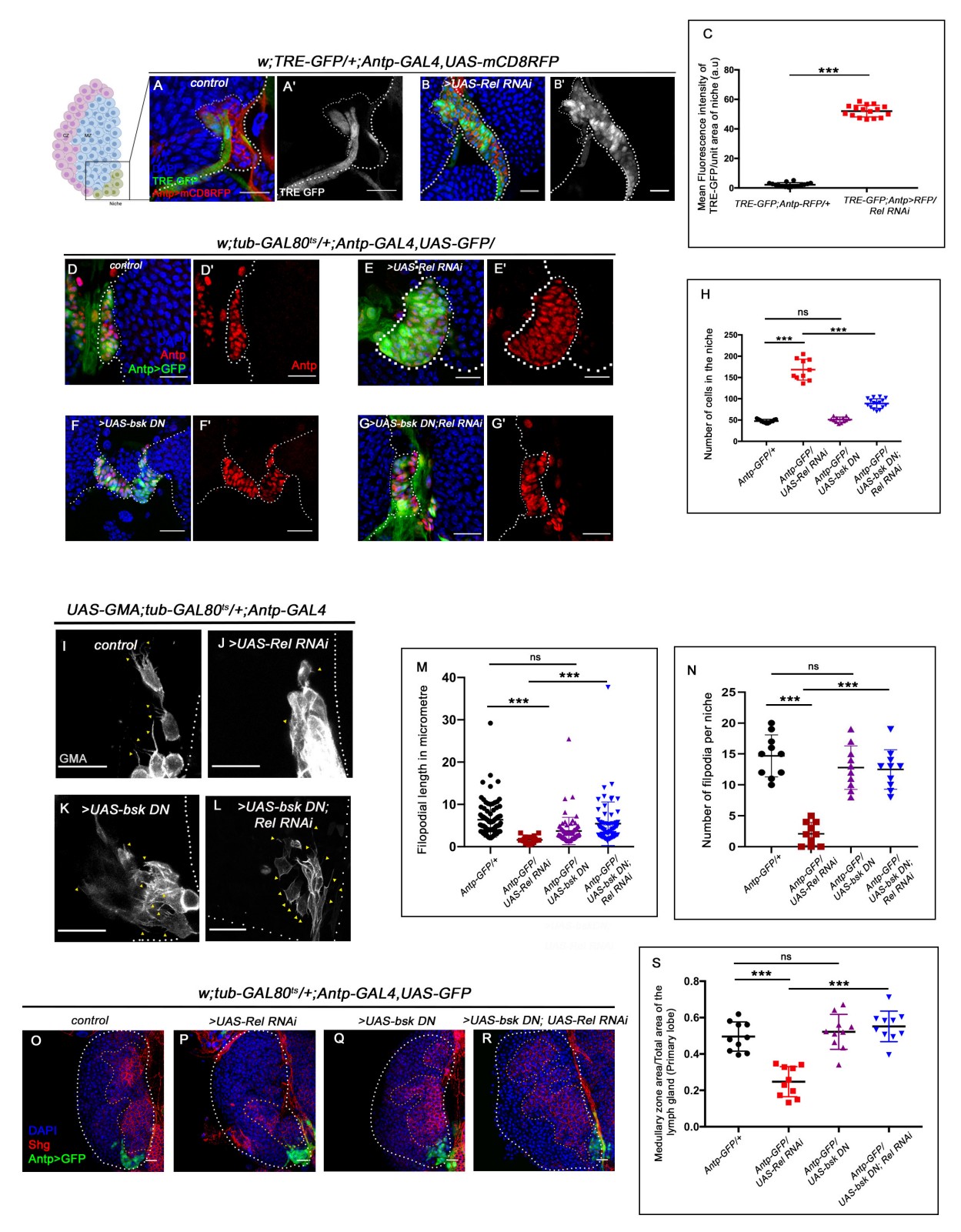

**Figure 5.** Loss of Relish from the niche activated JNK causing niche hyperplasia. The genotypes are mentioned in relevant panels. Scale bar: 20 µm. (A–B') Upregulation of JNK signaling visualized by its reporter TRE-GFP (green) in Relish knockdown (B–B') compared with WT niche (A–A'). (C) Statistical analysis of fluorescence intensity (A–B') revealed a significant increase in TRE-GFP levels compared to control (n=15, p-value=4.2 × 10$^{-19}$, two-tailed Student's t-test). (D–G') Upon niche-specific simultaneous knockdown of Rel and JNK, the niche hyperplasia observed upon loss of Relish (E–E') is

*Figure 5 continued on next page*

*Figure 5 continued*

rescued (**G–G'**) and is comparable to control (**D–D'**) whereas loss of bsk from the niche does not alter niche cell number (**F–F'**). (**H**) Statistical analysis of the data in (**D–G'**) (n=10, p-value=$5.6 \times 10^{-8}$ for control versus *Rel RNAi*, p-value=$8.0 \times 10^{-7}$ for *bsk DN; Rel RNAi* versus *Rel RNAi*, p-value=0.10 control versus for *bsk DN*; two-tailed unpaired Student's t-test). (**I–N**) Cellular filopodia from the niche cells in Rel loss of function is found to be smaller in length and fewer in numbers (**J and M–N**). Simultaneous loss of both JNK using *bsk DN* and Relish (**L and M–N**) rescued the stunted, scanty filopodia to control state (**I and M–N**), whereas loss of JNK did not affect filopodia formation (**K and M–N**). (**M–N**) Statistical analysis of the data in (**I–L**) (Filopodia number: n=10, p=$6.96 \times 10^{-8}$ for control versus *Rel RNAi*, p-value=$8.11 \times 10^{-7}$ for *bsk DN; Rel RNAi* versus *Rel RNAi*, p-value=0.153 for *bsk DN* versus control. Filopodia length: n=6, p-value=$2.78 \times 10^{-16}$ for control *versus Rel RNAi*, p-value=$1.84 \times 10^{-6}$ for *bsk DN; Rel RNAi* versus *Rel RNAi*, p-value=0.22 for *bsk DN* vs control; two-tailed unpaired Student's t-test). (**O–R**) Knocking down JNK function from the niche did not have any effect on progenitors (visualized by Shg) (**Q**). Downregulating bsk function in Rel loss genetic background was able to restore the reduction in prohemocyte pool (**R**) observed in Relish loss (**P**) scenario in comparison to control (**O**). (**S**) Statistical analysis of the data in (**O–R**) (n=10, p-value=$2.26 \times 10^{-6}$ for control versus *Rel RNAi*, p-value=$1.94 \times 10^{-7}$ for *bsk DN; Rel RNAi* versus *Rel RNAi*, p-value=0.521 for control versus *bsk DN*; two-tailed unpaired Student's t-test) The white dotted line marks whole of the lymph gland in all cases and niches in (**A–G'**). Yellow dotted lines mark the progenitor zone in (**O–R**). In all panels, age of the larvae is 96 hr AEH. The nuclei are marked with DAPI (blue). Individual dots represent biological replicates. Error bar: standard deviation (SD). Data are mean ± SD. *p<0.05, **p<0.01, and ***p<0.001.

The online version of this article includes the following source data and figure supplement(s) for figure 5:

**Source data 1.** Contains numerical data plotted in *Figure 5A–B'*, *Figure 5D–G'*, *Figure 5I–L*, *Figure 5O–R*, *Figure 5—figure supplement 1A–B'*, *Figure 5—figure supplement 1D–E'*, *Figure 5—figure supplement 1G–H''*, *Figure 5—figure supplement 1J–M*, *Figure 5—figure supplement 2A–D*, *Figure 5—figure supplement 2F–H'*, *Figure 5—figure supplement 3E–H*, *Figure 5—figure supplement 3J–M* and *Figure 5—figure supplement 3O–R*.

**Figure supplement 1.** Ectopic activation of JNK signaling in the niche affects niche cell proliferation and progenitor maintenance.

**Figure supplement 2.** Downregulating JNK in Relish loss genetic background rescues progenitor loss and precocious differentiation.

**Figure supplement 3.** Relish inhibits JNK signaling by restricting *tak1* activity in the niche during development.

number, as well as the proper communication between niche cells and progenitors, which is mandatory for the maintenance of the latter.

Collectively, these results indicate that Relish functions in the niche to repress JNK signaling during development. In the absence of this regulation, upregulated JNK causes cytoskeletal re-arrangements within the niche and disrupts Hh delivery to the progenitors. The morphogen trapped within the niche is unable to reach the progenitors, thereby affecting their maintenance.

## Relish inhibits JNK signaling by restricting *tak1* activity in the niche during development

It is essential to understand how the repression of JNK by Relish is brought about in a developmental scenario. Several in vitro and in vivo studies in vertebrates have shown the inhibitory role of NF-κB signaling over JNK during various developmental and immune responses (*Clark and Coopersmith, 2007*; *Nakano, 2004*; *Tang et al., 2001*; *Volk et al., 2014*). In *Drosophila*, mammalian MAP3 kinase homolog TAK1 activates both the JNK and NF-κB pathways following immune stimulation (*Boutros et al., 2002*; *Kaneko et al., 2006*; *Vidal et al., 2001*). Interestingly, during bacterial infection, Relish, once activated, leads to proteasomal degradation of TAK1, thereby limiting JNK signaling to prevent hyper-immune activation (*Park et al., 2004*). It is intriguing to speculate that a similar circuit is engaged in the niche to curtail JNK signaling during development. If this is the case, then the loss of *tak1* should restore the elevated TRE-GFP expression in a niche where Relish is downregulated. Indeed, upon genetic removal of one copy of *tak1* in conjunction with Relish loss from the niche, a drastic decrease in TRE-GFP expression is noted (*Figure 5—figure supplement 3A–D*). Furthermore, we found a significant reduction in cell number; analogous to what we observe when JNK and Relish activity is simultaneously downregulated from the niche (*Figure 5—figure supplement 3E–H and I*). It is interesting to note that there is a restoration in the progenitors (*Figure 5—figure supplement 3j–M and N*) along with the rescue of the precocious differentiation (*Figure 5—figure supplement 3O–R and S*) observed upon Relish loss from the niche, which is comparable to the control state in the above genotype.

These results led us to infer that Relish restricts the activation of JNK signaling in the hematopoietic niche via *tak1* during development. The restraint on JNK activity is essential for proper communication between niche cells and progenitor cells, which is necessary for maintaining the latter.

## Ecdysone-dependent activation of Relish in the niche is a developmental requirement

Cleavage, activation, and nuclear translocation of Relish during bacterial infection are brought about by binding of the cell wall component of gram-negative bacteria to membrane-bound receptor PGRP-LC (*Kaneko et al., 2006*; *Leulier et al., 2003*). We wondered whether the niche is employing a similar mechanism to regulate Relish activation during development by engaging the endogenous microbiota. To explore this possibility, we checked the status of the hematopoietic niche in the germ-free/axenic larvae (which were devoid of commensal microflora, *Figure 6—figure supplement 1A–A′ and B*). We found no significant change in the niche cell number in an axenic condition (*Figure 6A–B′ and D*) compared to the control. Additionally, JNK signaling (visualized by TRE-GFP) is not active in the hematopoietic niche of the axenic larva (*Figure 6—figure supplement 1C-C′*), neither the ectopic differentiation (visualized by Hemolectin, green) of the progenitors was evident (*Figure 6—figure supplement 1D-D′*). Furthermore, we employed a deletion mutant allele of *PGRP-LB* (PGRP-LB delta). This gene codes for an amide that specifically degrades gram-negative bacterial peptidoglycan (PGN) (*Paredes et al., 2011*; *Zaidman-Rémy et al., 2006*). Even in this scenario, where the systemic PGN level is known to be elevated, there is no increase in the niche cell number (*Figure 6C–C′ and D*). The above results demonstrate that during development, Relish expression and activation in the hematopoietic niche are independent of the commensal microflora.

Interestingly, activation of the IMD pathway components PGRP-LC and Relish is transcriptionally regulated by steroid hormone 20-hydroxyecdysone signaling during bacterial infection (*Rus et al., 2013*). Moreover, a recent study also reveals that the activation of Relish and IMD-dependent genes is mediated via ecdysone signaling in the Malpighian tubules during development (*Verma and Tapadia, 2015*). Strong expression of the Ecdysone receptor in the hematopoietic niche (*Figure 6—figure supplement 1E–E′′*) prompted us to check the possibility of ecdysone-dependent regulation of Relish expression and activation in the niche. Upon expression of a dominant-negative allele of the receptor EcR in the niche, a drastic reduction in the amount of Relish protein is evident (*Figure 6E–G′*). Intensity analysis reveals a threefold decrease in Relish expression upon blocking ecdysone signaling compared to the control niches (*Figure 6H*). Since transcriptional regulation of Relish through ecdysone signaling has been previously reported (*Rus et al., 2013*), we decided to explore whether this holds in case of the hematopoietic niche. Fluorescent in situ hybridization (FISH) analysis reveals the presence of *Rel* transcript in the lymph gland as well as in the salivary gland of control third-instar larvae (*Figure 6—figure supplement 2A–A′′ and C–C′*). Due to increase in differentiation, the number of *Rel* expressing progenitors is less compared to control (*Figure 6—figure supplement 2B–B′*). The sense probe was used as the negative control (*Figure 6—figure supplement 2D–E*).

To probe the status of Rel transcripts specifically in the niche, we performed whole-mount immunofluorescence (IF) along with FISH on the third-instar lymph gland. Drastic reduction of the *Rel* transcript is noticeable in the niche from where EcR expression was downregulated compared to the control (*Figure 6—figure supplement 2F–G′′ and H*), implicating that *Rel* is transcriptionally regulated through ecdysone signaling.

This observation indicates that the phenotypes observed upon EcR loss from the niche should be analogous to Rel loss. Attenuation of ecdysone signaling indeed leads to a significant increase in niche cell proliferation compared to the control (*Figure 6I–K′ and L*). Furthermore, to understand whether the functionality of the niche is also compromised in the above genotype, we checked the differentiation status. Similar to Relish loss, downregulation of ecdysone signaling from the niche results in precocious differentiation (*Figure 6M–O′ and P*). Niche-specific overexpression of Rel in conjunction with EcR loss can restore the cell number of the niche (*Figure 6Q–T′ and U*) as well as its functionality (*Figure 6—figure supplement 2I–L and M*).

These results, therefore, collectively suggest that ecdysone signaling regulates the expression and activation of Relish in the hematopoietic niche during development (*Figure 6—figure supplement 2N*). These results also underscore the requirement of a hormonal signal in regulating Relish during developmental hematopoiesis.

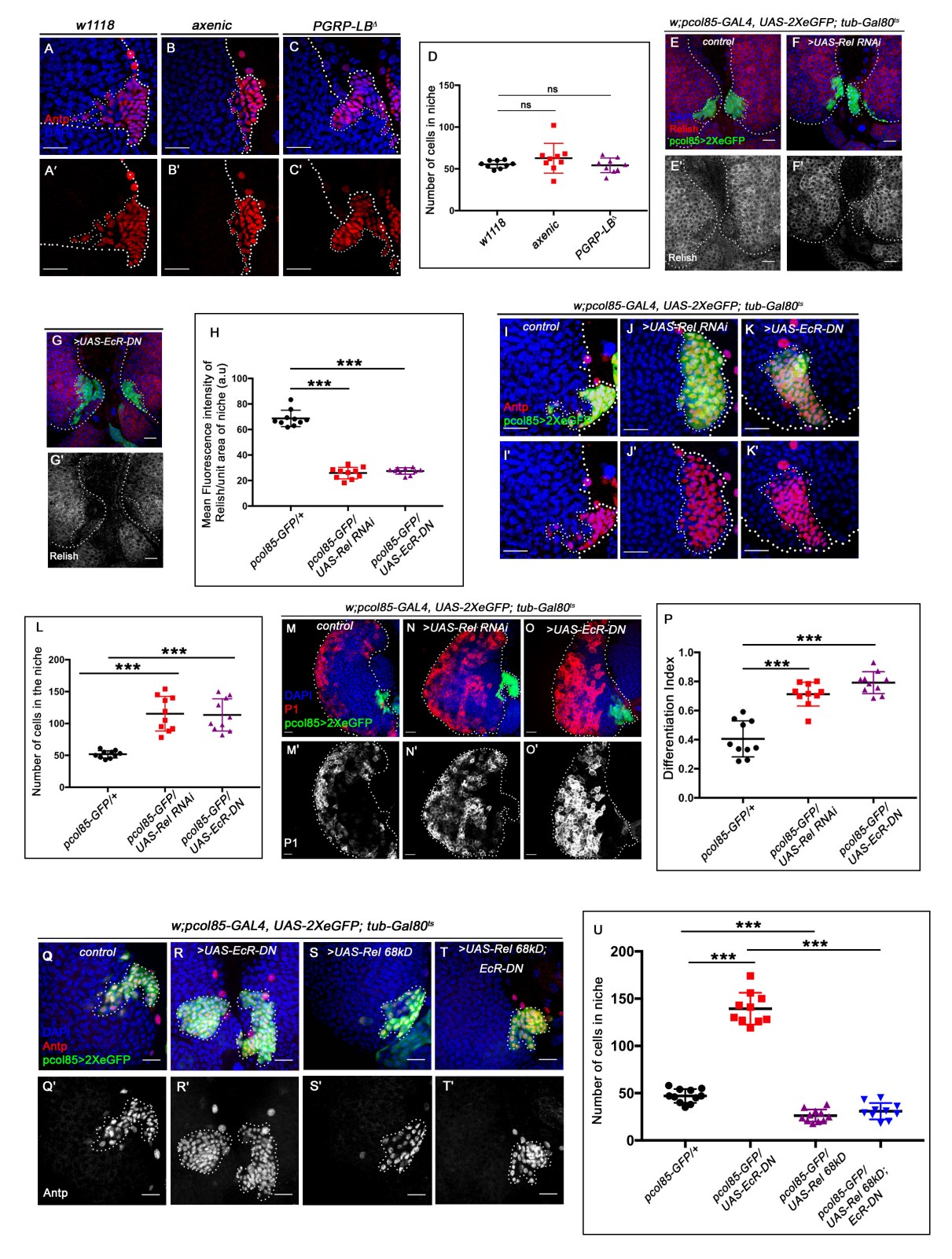

**Figure 6.** Ecdysone regulates Relish expression and functionality in the niche. The genotypes are mentioned in relevant panels. Scale bar: 20 μm. (A–C′) Niche number remains comparable to control (A–A′) both in axenic larval lymph gland (B–B′) and in PGRP-LB mutant where there is upregulation in systemic peptidoglycan levels (C–C′). (D) Statistical analysis of the data in (A–C′) (n=9; p-value = 0.262 for control versus germ free and 0.392 for control versus PGRP-LB mutant; two-tailed unpaired Student's t-test). (E–G′) Compared to that of control (E–E′) Rel expression is significantly downregulated

*Figure 6 continued on next page*

*Figure 6 continued*

both in EcR loss (**G–G'**) as well as in Rel loss from the niche (**F–F'**). (**H**) Statistical analysis of the data in (**E–G'**) (n=10, p-value=7.81 × 10$^{-12}$ for control versus *Rel RNAi* loss and p-value = 3.76 × 10$^{-10}$ for control versus *EcR-DN*; two-tailed unpaired Student's t-test). (**I–K'**) Similar to Rel loss from the niche (**J–J'**), EcR loss also results in increase in niche cell numbers (**K–K'**) compared to that of control (**I–I'**). (**L**) Statistical analysis of the data in I-K' (n=10, p-value=6.6 × 10$^{-5}$ for control versus *EcR-DN* and p-value = 3.1x10$^{-5}$ for control versus *Rel RNAi*; two-tailed unpaired Student's t-test). (**M–O'**) Compared to control (**M–M'**), both loss of Rel (**N–N'**) and EcR (**O–O'**) from the niche results in increase in differentiation. (**P**) Statistical analysis of the data in (**M–O'**) (n=10, p-value=4.3 × 10$^{-5}$ for control versus *Rel RNAi* and p-value=2.2 × 10$^{-6}$ for control versus *EcR-DN*; two-tailed unpaired Student's t-test). (**Q–T'**) Increase in niche cell numbers observed upon EcR loss from the niche (**R–R'**) is rescued to control levels (**Q–Q'**) when Relish was overexpressed in an EcR loss genetic background (**T–T'**). Overexpression of Relish in the niche reduced the cell number compared to control (compare **S–S'** and **Q–Q'**). (**U**) Statistical analysis of the data in (**Q–T'**) (n=10; p-value=1.7×10$^{-9}$ for control versus *EcR-DN*, p-value=7.8 × 10$^{-11}$ for *Ecr-DN* versus *UAS-Rel 68kD; EcR-DN*, p-value=3.63 × 10$^{-6}$ for control versus *UAS-Rel 68kD*; two-tailed unpaired Student's t-test). The white dotted line marks whole of the lymph gland and niches in all the cases. In all panels, age of the larvae is 96 hr AEH. The nuclei are marked with DAPI (blue). Individual dots represent biological replicates. Error bar: standard deviation (SD). Data are mean ± SD. *p<0.05, **p<0.01, and ***p<0.001.

The online version of this article includes the following source data and figure supplement(s) for figure 6:

**Source data 1.** Contains numerical data plotted in *Figure 6A–C'*, *Figure 6E–G'*, *Figure 6I–K'*, and *Figure 6M–O'*, *Figure 6Q–T'*, *Figure 6—figure supplement 2F–G''*, *Figure 6—figure supplement 2I–L*.

**Figure supplement 1.** Ecdysone signaling is active in the hematopoietic niche.

**Figure supplement 2.** Relish expression is transcriptionally regulated by ecdysone signaling in the hematopoietic niche.

## During bacterial infection Relish in the niche is downregulated to facilitate immune response

In *Drosophila*, ecdysone-mediated immune potentiation has shown to have a greater impact on the development of immunity in embryos (*Tan et al., 2014*) as well as the survival of flies during bacterial infection (*Flatt et al., 2008*; *Rus et al., 2013*; *Tan et al., 2014*; *Verma and Tapadia, 2015*; *Xiong et al., 2016*). Interestingly, we found a fourfold decrease in Relish expression from the hematopoietic niche during bacterial infection compared to uninfected larvae (compared *Figure 7A–A'* with C -C' and quantitated in *Figure 7D*). To rule out the possible effect of injection on Rel expression, we compared the infected with sham control. There was a 2.6-fold decrease in the intensity of Rel expression within the niche of infected larvae compared to the sham control (compare *Figure 7B–B'* with C–C', quantitated in *Figure 7D*). In contrast, upon bacterial infection, we could see the nuclear expression of Relish in the fat body cells as previously reported (*Figure 7E–G*; *Cha et al., 2003*; *Kim et al., 2006*). Interestingly, niche-specific overexpression of the N-terminal domain of Relish (*UAS-Rel68kD*), which is known to translocate to the nucleus and induce target gene expression (*Stöven et al., 2000*), is unable to sustain Relish expression post-infection (*Figure 7H–H'*), implicating the post-transcriptional regulation on Relish during bacterial infection. Relish activity is modulated through proteasomal degradation in *Drosophila* and *Bombyx mori* (*Khush et al., 2002*; *Ma et al., 2015*).

More importantly, we also found that compared to control, 4 hr post-bacterial challenge, the progenitor pool declines (*Figure 7I–K*), accompanied by a concomitant precocious differentiation (*Figure 7L–N*). These phenotypes show a remarkable similarity to the ones seen on the loss of Relish from the niche (*Figure 1H–P*). As a response to systemic bacterial infection, upregulation of JNK is detected throughout the lymph gland, including the niche compared to sham control (*Figure 7—figure supplement 1A–B'*). The short duration of systemic infection adopted in our study induced proliferation in the otherwise quiescent niche cells (*Figure 7—figure supplement 1C–E*). Based on these studies, we speculate that Relish, in this case, might also undergo ubiquitin-mediated degradation (by Factor X, *Figure 7—figure supplement 1F*) that overrides the developmental signal (*Figure 6—figure supplement 2M*) during bacterial infection.

These data collectively elucidate that a differential regulation on Relish is mandatory during bacterial infection to boost immune response.

## Discussion

Our study unravels the molecular genetic basis of the hormonal control on Relish expression in the hematopoietic niche essential for maintaining the hemocyte progenitors of the lymph gland during development. Hemocytes present in the lymph gland are not actively involved in immune

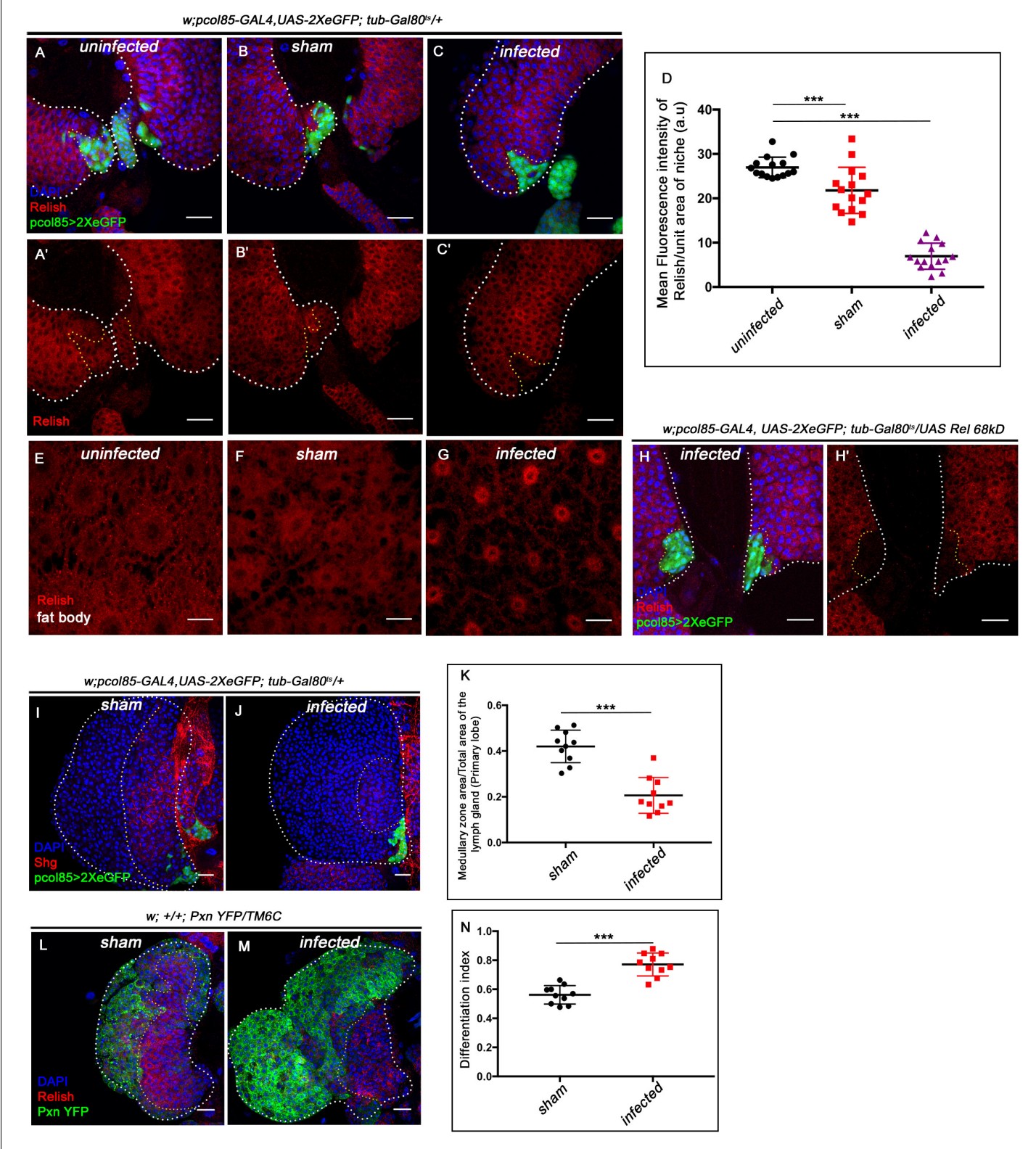

**Figure 7.** Niche-specific expression and function of Relish is susceptible to pathophysiological state of the organism. The genotypes are mentioned in relevant panels. Scale bar: 20 μm. (A–C') Compare to uninfected conditions (A–A') and sham (B–B'), significant reduction in Relish expression was observed in the hematopoietic niche 4 hr post-infection (C–C'). (D) Statistical analysis of the data in (A–C') (n=15; p=6.62×10⁻¹⁸ for unpricked versus infected, p=2.5×10⁻⁷ for sham versus infected, two-tailed unpaired Student's t-test). (E–G) Nuclear expression of Relish was observed in infected (G) fat

*Figure 7 continued on next page*

*Figure 7 continued*

body cells 4 hr post in contrast to uninfected (E) and sham (F) larval fat body. (H–H') Overexpressing Relish N-terminus (*UAS-Rel-68kD*) could not rescue loss of Relish expression post-infection. (I–J) Compared to sham (I), significant reduction in Shg-positive progenitors (red) were observed in infected lymph glands (J). (K) Statistical analysis of the data in (I–J) (n=10; p-value=5.2 × 10$^{-6}$ for sham versus infected, two-tailed unpaired Student's t-test). (L–M) Drastic increase in differentiation (visualized by *Pxn-YFP*, green) was observed in infected lymph glands (M) compared to sham (L). (N) Statistical analysis of the data in (L-M) (n=10; p-value = 4.65×10$^{-6}$ for sham versus infected, two-tailed unpaired Student's t-test). The white dotted line mark whole of the lymph gland in all cases. Yellow dotted line marks the niche in (A– C' and H–H') and the boundary between CZ and MZ in (L–M). In all panels, age of the larvae is 96 hr AEH. The nuclei are marked with DAPI (Blue). Individual dots represent biological replicates. Error bar: standard deviation (SD). Data are mean ± SD. *p<0.05, **p<0.01, and ***p<0.001.

The online version of this article includes the following source data and figure supplement(s) for figure 7:

**Source data 1.** Contains numerical data plotted in *Figure 7A–C'*, *Figure 7I–J*, *Figure 7L–M*, *Figure 7—figure supplement 1C–D*.
**Figure supplement 1.** Upregulation in JNK signaling and increase in cell proliferation was observed in the niche during infection.

surveillance under healthy conditions. Within this organ, the hemocytes proliferate to create a pool of progenitors and differentiated cells. However, with its content, this organ takes care of all post-larval hematopoiesis and therefore is not precociously engaged. Our study illustrates how the hematopoietic niche recruits neuroendocrine-immunity (Ecdysone–Relish) axis to maintain the progenitors of the lymph gland during larval development (*Figure 6—figure supplement 2M*). The loss of Ecdysone/Relish, therefore, results in precocious maturation of the progenitors. The mechanism underlying the control of niche state and function by Relish involves repression of the Jun Kinase signaling. Interestingly, Relish during infection is known to inhibit JNK activation in response to gram-negative bacterial infection in *Drosophila* (*Park et al., 2004*). We found that this antagonistic relation of Relish and JNK, essential for innate immunity, is also relevant during development to facilitate the functioning of the hematopoietic niche. Our results suggest that two independent events occur in the niche if JNK is activated (*Figure 6—figure supplement 2M*). Firstly, the activation of JNK leads to supernumerary niche cells due to an increase in Wingless expression. Secondly, the JNK pathway negatively regulates the actin-based cytoskeletal architecture essential for the release of Hh from the niche cells.

Though perceived as a pro-apoptotic signal, a large body of work has evidenced the role of the JNK pathway to induce proliferation in diverse developmental scenarios (*Kaur et al., 2019*; *Ohsawa et al., 2012*; *Pérez-Garijo et al., 2009*; *Pinal et al., 2019*; *Wu et al., 2010*). The JNK pathway is also known for its ability to release proliferative signals that can stimulate the growth of the tissue (*Pinal et al., 2019*). For instance, during compensatory proliferation in the developing larval wing disc, JNK triggers wingless to stimulate the proliferation of the non-dead cells (*Ryoo et al., 2004*). Moreover, wingless signaling has been reported as a mitogenic signal for stem cells in diverse contexts (*Deb et al., 2008*; *Lin et al., 2008*; *Song and Xie, 2003*), and aberrant activation of this pathway contributes to various blood cell disorders and cancers (*Grainger and Willert, 2018*; *Klaus and Birchmeier, 2008*; *Lento et al., 2013*; *Reya and Clevers, 2005*). *Drosophila* hematopoietic niche is known to positively rely upon wingless (Wg) signaling for its proliferation during larval development. Downregulation of the signaling by expressing a dominant-negative form of its receptor Frizzled results in a reduction in niche cell numbers (*Sinenko et al., 2009*). We believe, to prevent hyperproliferation of the niche cells, Relish is reining in Wingless by inhibiting JNK signaling during development.

Several studies have shown that actin-based cellular extensions or cytonemes (*Bischoff et al., 2013*; *González-Méndez et al., 2019*; *Gradilla et al., 2014*; *Kornberg and Roy, 2014*; *Portela et al., 2019*) play a crucial role in transporting Hh from the source to several cell diameter distances (*Rojas-Ríos et al., 2012*), thereby contributing in the establishment of Hh gradient. Coincidently, *Drosophila* hematopoietic niche cells are also known to emanate cytoneme-like filopodial projections to the nearby progenitor cells (*Mandal et al., 2007*; *Pennetier et al., 2012*; *Tokusumi et al., 2011*). We demonstrate that perturbation of this filopodial extension disrupts the transportation of Hh from the niche. The current study is in sync with the understanding that these cellular extensions are required to maintain the undifferentiated cell population by facilitating the crosstalk between niche and hematopoietic progenitors (*Krzemień et al., 2007*; *Tokusumi et al., 2011*). Here, we show that upon Relish loss from the niche, filopodial formation gets impaired in a JNK- dependent manner. Ectopic activation of JNK signaling leads to altered expression of

cytoskeletal elements that disrupt the process of filopodial formation. Consequently, the morphogen Hh gets trapped within the niche cells, thereby hamper the proper communication between the niche and the progenitor cells of the lymph gland (*Figure 8*). Previous studies have demonstrated that activation of Relish leads to the disruption of cytoskeletal architecture in S2 cells to bring about the necessary changes associated with cell shapes for the proper immune response (*Foley and O'Farrell, 2004*). However, the underlying mechanism of the modulation of cytoskeletal elements by Relish was not evident. Here we provide in vivo genetic evidence for the process by which Relish loss causes alteration of the cytoskeletal elements of the niche cells by ectopic JNK activation.

Another enthralling finding of our study is identifying 20-hydroxyecdysone signaling as a regulator of *Drosophila* developmental hematopoiesis. The underlying reasons for this hormonal control on Relish seem to be intriguing. The need for this regulatory network during development may be related to the various microbial threats commonly confronted and dealt with by the circulating hemocytes of the larvae. While the circulating hemocytes cater to this need, the blood cells in the lymph gland proliferate and undergo maturation, creating a reservoir of hemocytes dedicated to deal with the post-larval requirements. Therefore, to safeguard the reserve population from responding to all of the common threats faced during development, the niche employs the ecdysone–Relish axis to prevent the disruption in definitive hematopoiesis. However, during a high infection load, the lymph gland ruptures, suggesting a break in this circuit. This notion gets endorsed when the niche is analyzed post-infection. A previous study demonstrated that the septate junction

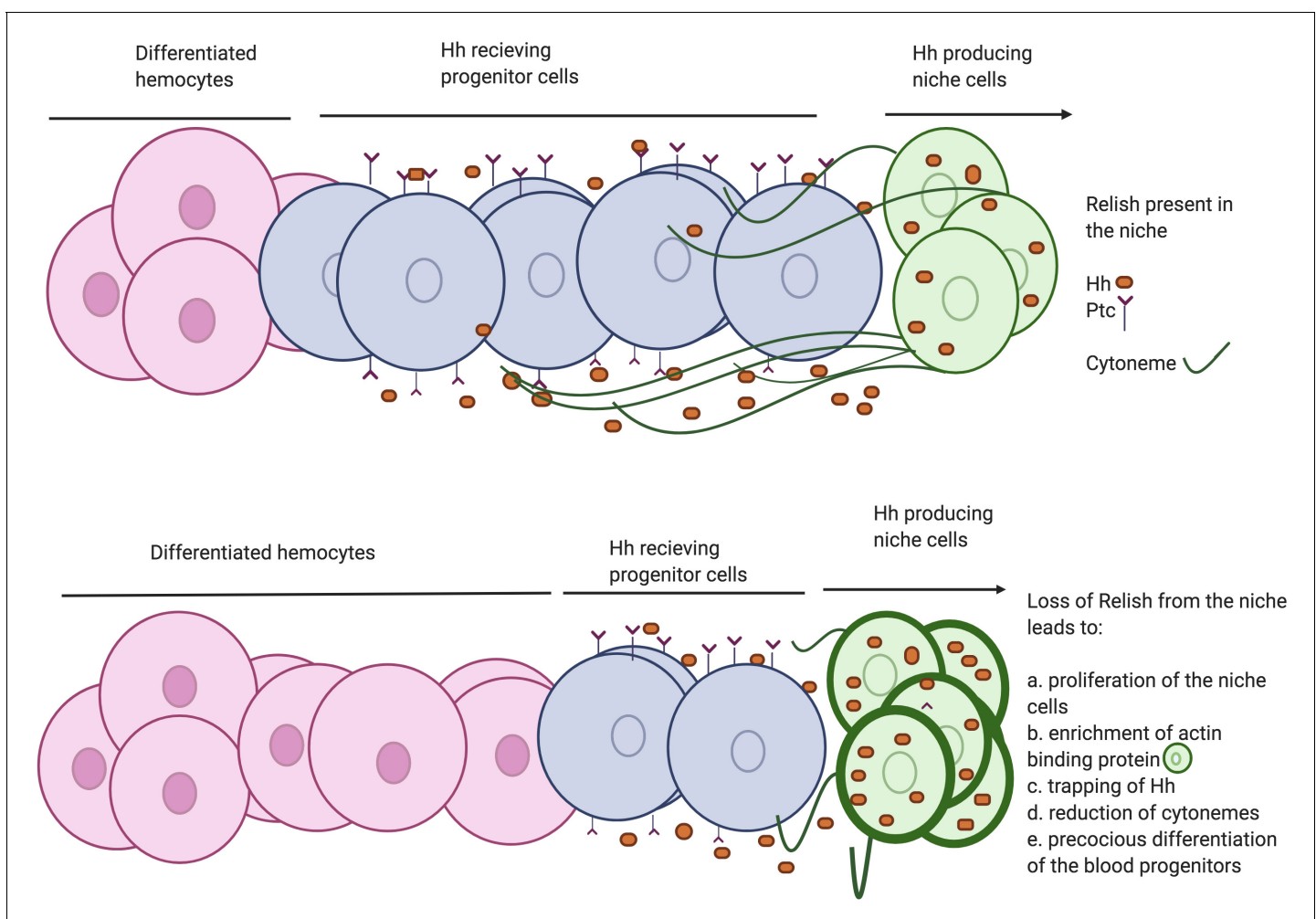

**Figure 8.** Developmental requirement of Relish in the niche for progenitor maintenance. Scheme describing how loss of Relish from the niche alters cytoskeletal elements of the cells. The change in cytoskeletal architecture affects cytoneme-like filopodial formation thereby trapping Hedgehog within the niche. The failure of Hh delivery in turn interferes with progenitor maintenances and pushes them toward differentiation.

in the niche is dismantled during infection, leading to the disbursing of differentiation signals that facilitated the maturation of the hemocytes (*Khadilkar et al., 2017*). We demonstrate that bacterial infection results in downregulation of Rel from the niche, which alters cytoskeletal architecture and traps the maintenance signal. As a consequence, precocious differentiation sets in the lymph gland, while in the case of the earlier study, seeping out of too many differentiation signals leads to ectopic differentiation underlining the fact that maintenance and differentiation are both sides of the same coin.

Quite intriguingly, the downregulation of Relish in the niche during bacterial infection and the response of the lymph gland mimic the genetic loss of Relish from the niche. These observations confirm that the developmental pathway gets tweaked in the hematopoietic niche to combat high bacterial infection (*Figure 7—figure supplement 1F*).

During bacterial infection, the activation of Relish by ecdysone signaling in the fat body results in the production of antimicrobial peptides (*Rus et al., 2013*). In contrast to this, we show that upon infection, Relish needs to be downregulated in the niche to bolster the cellular immune response. This downregulation of Relish facilitates the release of a large pool of macrophages from the lymph gland to augment the circulating hemocytes to combat infection. The lymph gland hemocytes do not participate in immune surveillance during development. However, during wasp infection, activation of the Toll/NF-κB signaling occurs in the niche to recruit lymph gland hemocytes to encapsulate wasp eggs (*Louradour et al., 2017*). We show that during bacterial infections, Relish, another member of the NF-κB pathway, is downregulated in the niche to disperse the lymph gland hemocytes into circulation. It is intriguing to see that the contrasting regulation of NF-κB components by the hematopoietic niche is essential for mounting an adequate immune response.

Interestingly, de novo production of neutrophils occurs in the bone marrow in response to systemic bacterial infection (*Zhao and Baltimore, 2015*). In mouse, 'emergency granulopoiesis' demands the activation of the TLR (Toll-like receptors)/NF-κB pathway via TLR4 in the vascular niche (*Boettcher et al., 2014*). It will be important to investigate whether this differential regulation on NF-κB members is evident in vertebrate bone marrow niches during infection.

For an organism to combat an infection successfully, a quick shift of the ongoing hematopoiesis toward emergency mode is absolutely necessary. We show that the hematopoietic niche is the sensor that gauges the physiological state of the animal and diverts the basal hematopoiesis toward the emergency hematopoiesis.

In conclusion, the present work reveals an unexpected role of Relish in developmental hematopoiesis. Furthermore, it unravels the systemic regulation of the hematopoietic niche by the neuroendocrine system. Also, it sheds light on how, during infection, this pathway gets suppressed to reinforce the cellular arm of the innate immune response.

# Materials and methods

**Key resources table**

| Reagent type (species) or resource | Designation | Source or reference | Identifiers | Additional information |
|---|---|---|---|---|
| Gene (*Drosophila melanogaster*) | Antp | Flybase:FB2020_01 | FLYB:FBgn0260642 | |
| Gene (*Drosophila melanogaster*) | Hml | Flybase:FB2020_01 | FLYB:FBgn 0029167 | |
| Gene (*Drosophila melanogaster*) | Collier/kn | Flybase:FB2020_01 | FLYB:FBgn0001319 | |
| Gene (*Drosophila melanogaster*) | wg | Flybase:FB2020_01 | FLYB: FBgn0284084 | |

*Continued on next page*

*Continued*

| Reagent type (species) or resource | Designation | Source or reference | Identifiers | Additional information |
|---|---|---|---|---|
| Gene (*Drosophila melanogaster*) | hep | Flybase:FB2020_01 | FLYB:FBgn0010303 | |
| Gene (*Drosophila melanogaster*) | EcR | Flybase:FB2020_01 | FLYB:FBgn0000546 | |
| Gene (*Drosophila melanogaster*) | PGRP-LB | Flybase:FB2020_01 | FLYB:FBgn0037906 | |
| Gene (*Drosophila melanogaster*) | Tak1 | Flybase:FB2020_01 | FLYB:FBgn0026323 | |
| Gene (*Drosophila melanogaster*) | bsk | Flybase:FB2020_01 | FLYB:FBgn 0000229 | |
| Gene (*Drosophila melanogaster*) | Ena | Flybase:FB2020_01 | FBgn0000578 | |
| Gene (*Drosophila melanogaster*) | Hh | Flybase:FB2020_01 | FBgn0004644 | |
| Gene (*Drosophila melanogaster*) | Dia | Flybase:FB2020_01 | FBgn0011202 | |
| Genetic reagent (*D. melanogaster*) | Antp-Gal4 | ***Emerald and Cohen, 2004*** | FLYB:FBal0155891 | FlyBase symbol: GAL4$^{Antp-21}$ |
| Genetic reagent (*D. melanogaster*) | P(*col5-cDNA*)/CyO-TM6B, Tb | ***Krzemień et al., 2007*** | FLYB:FBti0077825 | FlyBase symbol: P{GAL4}col85 |
| Genetic reagent (*D. melanogaster*) | Hml-GAL4.Δ | ***Sinenko and Mathey-Prevot, 2004*** | FLYB:FBtp0040877 | FlyBase symbol: P{Hml-GAL4.Δ} |
| Genetic reagent (*D. melanogaster*) | UAS-Rel RNAiKK | Vienna *Drosophila* Resource Center | VDRC:v108469; FLYB:FBti0116709; RRID:FlyBase_ FBst0477227 | FlyBase symbol: P{KK100935}VIE-260B |
| Genetic reagent (*D. melanogaster*) | w[1118] | Bloomington *Drosophila* Stock Center | BDSC:3605; FLYB:FBal0018186; RRID:BDSC_3605 | FlyBase symbol: w$^{1118}$ |
| Genetic reagent (*D. melanogaster*) | UAS-Rel RNAi | Bloomington *Drosophila* Stock Center | BDSC:33661; FLYB:FBti0140134; RRID:BDSC33661 | FlyBase symbol: P{TRiP. HMS00070}attP |
| Genetic reagent (*D. melanogaster*) | UAS-wg RNAi | Bloomington *Drosophila* Stock Center | BDSC:33902; FLYB:FBal0263076; RRID:BDSC_33902 | FlyBase symbol: P{TRiP. HMS00844}attP2 |
| Genetic reagent (*D. melanogaster*) | UAS-dia RNAi | Bloomington *Drosophila* Stock Center | BDSC:35479; FLYB:FBtp0068562; RRID:BDSC_35479 | FlyBase symbol: P{TRiP.GL00408} |
| Genetic reagent (*D. melanogaster*) | UAS-hep.Act | Bloomington *Drosophila* Stock Center | BDSC:9305; FLYB:FBti0074410; RRID:BDSC_9305 | FlyBase symbol: P{UAS-Hep.Act}1 |
| Genetic reagent (*D. melanogaster*) | UAS-FUCCI | Bloomington *Drosophila* Stock Center | BDSC:55121; RRID:BDSC_55121 | FlyBase symbol: P{UAS-GFP.E2f1.1–230}32; P{UAS-mRFP1.NLS.CycB. 1–266}19 |
| Genetic reagent (*D. melanogaster*) | TRE-GFP | Bloomington *Drosophila* Stock Center | BDSC:59010; FLYB:FBti0147634; RRID:BDSC_59010 | FlyBase symbol: P{TRE-EGFP}attP16 |

*Continued on next page*

*Continued*

| Reagent type (species) or resource | Designation | Source or reference | Identifiers | Additional information |
|---|---|---|---|---|
| Genetic reagent (*D. melanogaster*) | Pxn-YFP | Kyoto Stock Center | kyoto:115452; FLYB: FBti0143571; RRID:FlyBase_ FBst0325439 | FlyBase symbol: PBac{802 .P.SVS-2} Pxn[CPTI003897] |
| Genetic reagent (*D. melanogaster*) | hhF4f-GFP | *Tokusumi et al., 2012* | FBtp0070210 | FlyBase symbol:P{hhF4f-GFP} |
| Genetic reagent (*D. melanogaster*) | UAS-GMA | Bloomington *Drosophila* Stock Center | BDSC:31774; FLYB:FBti0131130; RRID:BDSC_31774 | FlyBase symbol:P{UAS-GMA}1 |
| Genetic reagent (*D. melanogaster*) | UAS-Rel 68kD | Bloomington *Drosophila* Stock Center | BDSC:55778; FLYB:FBti0160486; RRID:BDSC_55778 | FlyBase symbol: P{UAS-FLAG-Rel.68}i21-B |
| Genetic reagent (*D. melanogaster*) | UAS-Rel 68kD | Bloomington *Drosophila* Stock Center | BDSC:55777; FLYB:FBti0160484 RRID:BDSC_55777 | FlyBase symbol: P{UAS-FLAG-Rel.68} |
| Genetic reagent (*D. melanogaster*) | UAS-EcR.B1Δ | Bloomington *Drosophila* Stock Center | BDSC:6872; FLYB:FBti0026963; RRID:BDSC_6872 | FlyBase symbol: P{UAS-EcR.B1-ΔC655.W650A}TP1-9 |
| Genetic reagent (*D. melanogaster*) | PGRP-LB[Delta] | Bloomington *Drosophila* Stock Center | BDSC:55715; FLYB:FBti0180381; RRID:BDSC_55715 | FlyBase symbol: TI{TI}PGRP-LB[Δ] |
| Genetic reagent (*D. melanogaster*) | wgl-12 cn1 bw1/CyO | Bloomington *Drosophila* Stock Center | BDSC:7000; FLYB:FBal0018504; RRID:BDSC_7000 | FlyBase symbol: wg[l-12] |
| Genetic reagent (*D. melanogaster*) | Tak1(2) | Bloomington *Drosophila* Stock Center | BDSC:26272; FLYB:FBal0131420; RRID:BDSC_26272 | FlyBase symbol: dTak1[2] |
| Gene (*Drosophila melanogaster*) | Rel[E20] | Flybase: FB2020_01 | FLYB:FBgn0014018 | |
| Genetic reagent (*D. melanogaster*) | UAS-bsk[DN] | Bloomington *Drosophila* Stock Center | BDSC:6409; FLYB:FBti0021048; RRID:BDSC_6409 | FlyBase symbol: P{UAS-bsk.DN}2 |
| Genetic reagent (*D. melanogaster*) | UAS-ena RNAiKK | Vienna *Drosophila* Resource Center | VDRC: v106484 FBst0478308; RRID:v106484 | FlyBase symbol: P{KK1077 52}VIE-260B |
| Genetic reagent (*D. melanogaster*) | UAS-mCD8: RFP | Bloomington *Drosophila* Stock Center | BDSC:27400; FLYB:FBti0115747; RRID:BDSC_27400 | FlyBase symbol: P{UAS-mCD8. mRFP.LG}28a |
| Genetic reagent (*D. melanogaster*) | tubGAL80 [ts20] | Bloomington *Drosophila* Stock Center | BDSC:7109; FLYB:FBti0027796; RRID:BDSC_7109 | FlyBase symbol: P{tubP-GAL80[ts]}20 |
| Antibody | Anti-P1 (Mouse monoclonal) | *Kurucz et al., 2007* | Cat# NimC1, RRID:AB_2568423 | IF(1:50) |
| Antibody | Anti-c Rel (Mouse monoclonal) | *Stöven et al., 2000* | Cat#21F3, RRID:AB_1552772 | IF (1:50) |
| Antibody | Anti-Ci[155] (Rat polyclonal) | Developmental Studies Hybridoma Bank | Cat# 2A1, RRID:AB_2109711 | IF(1:2) |
| Antibody | Anti-Wg (Mouse monoclonal) | Developmental Studies Hybridoma Bank | Cat#4D4 RRID:AB_528512 | IF(1:3) |
| Antibody | Anti-Singed (Mouse monoclonal) | Developmental Studies Hybridoma Bank | Cat# sn 7C RRID:AB_528239 | IF(1:20) |

*Continued on next page*

*Continued*

| Reagent type (species) or resource | Designation | Source or reference | Identifiers | Additional information |
|---|---|---|---|---|
| Antibody | Anti-Enabled (Mouse monoclonal) | Developmental Studies Hybridoma Bank | Cat#5G2 RRID:AB_528220 | IF(1:30) |
| Antibody | Anti-PH3(Rabbit monoclonal) | Cell signaling Technology | Cat# 3642S RRID:AB_10694226 | IF(1:150) |
| Antibody | Anti-Hh (Rabbit monoclonal) | *Forbes et al., 1993* | | IF(1:500) |
| Antibody | Anti-Hnt (Mouse monoclonal) | Developmental Studies Hybridoma Bank | Cat#1G9 RRID:AB_528278 | IF(1:5) |
| Antibody | Anti-EcR common (Mouse monoclonal) | Developmental Studies Hybridoma Bank | Cat#DDA2.7 RRID:AB_10683834 | IF(1:20) |
| Antibody | Anti-Ance (rabbit monoclonal) | *Hurst et al., 2003* | | IF(1:500) |
| Antibody | Anti-GFP (rabbit polyclonal) | Cell signaling Technology | Cat#2555 | IF(1:100) |
| Antibody | Anti-shg (rat monoclonal) | Developmental Studies Hybridoma Bank | Cat#DCAD2 RRID:AB_528120 | IF(1:50) |
| Antibody | Anti-β-PS (mouse monoclonal) | Developmental Studies Hybridoma Bank | Cat#CF.6G11 RRID:AB_528310 | IF(1:3) |
| Antibody | Anti-DIG-POD (sheep polyclonal) | Sigma-Aldrich | Cat#11207733910 | IF(1:1000) |
| Chemical compound, drug | Phalloidin from *Amanita phalloides* | Sigma-Aldrich | Cat#P2141 | IF(1:500) |
| Chemical compound, drug | Rhodamine Phalloidin | Thermo Scientific | Cat# R415 RRID:AB_2572408 | IF(1:500) |
| Sequence-based reagent | Relish cDNA clone | DGRC | Clone id: GH01881 FLYB: FBcl0110737 | |
| Sequence-based reagent | Actin_F | *Elgart et al., 2016* | PCR primers | GGAAACCACGCA AATTCTCAGT |
| Sequence-based reagent | Actin_R | *Elgart et al., 2016* | PCR primers | CGACAACCAGA GCAGCAACTT |
| sequence-based reagent | Aceto_F | *Elgart et al., 2016* | PCR primers | TAGTGGCGGAC GGGTGAGTA |
| Sequence-based reagent | Aceto_R | *Elgart et al., 2016* | PCR primers | AATCAAACGCA GGCTCCTCC |
| Sequence-based reagent | Lacto_F | *Elgart et al., 2016* | PCR primers | AGGTAACGGCTC ACCATGGC |
| Sequence-based reagent | Lacto_R | *Elgart et al., 2016* | PCR primers | ATTCCCTACTGC TGCCTCCC |
| Software, algorithm | Fiji | Fiji | RRID:SCR_002285 | |
| Software, algorithm | Photoshop CC | Adobe | RRID:SCR_014199 | |
| Software, algorithm | Imaris | Bitplane | RRID:SCR_007370 | |
| Commercial assay or kit | Click-iTEdU plus (DNA replication kit) | Invitrogen | Cat# C10639 | |
| Commercial assay or kit | Alexa Fluor 594 Tyramide Reagent | Thermo Fischer | Cat# B40957 | |

## Fly stocks

In this study, the following *Drosophila* strains were used: *Antp-Gal4* (S. Cohen, University of Copenhagen, Denmark), *PCol85-Gal4* (M. Crozatier, Université de Toulouse, France), *Rel$^{E20}$* (B. Lemaitre, École polytechnique fédérale de Lausanne, Switzerland), and *hhF4f*-GFP (R. Schulz, University of Notre Dame, USA). *Hml-GAL4.Δ* (S. Sinenko, Russian Academy of Sciences, Moscow), *UAS-Rel RNAi* (II), *Pxn-YFP*, and *UAS-ena RNAi* (II) were from the Vienna *Drosophila* Resource Center. The following stocks were procured from Bloomington *Drosophila* Stock Center: w1118, *UAS-Rel RNAi*, *UAS-Rel 68kD* (I), *UAS-Rel 68kD* (II), *UAS-EcR.B1Δ*, *PGRP-LBΔ*, *UAS-wg RNAi*, *UAS-dia RNAi*, *TRE-GFP*, *UAS-bsk DN*, *UAS-mCD8-RFP*, *UAS-Hep$^{act}$*, *wg$^{ts}$/cyo*, *UAS-GMA*, *UAS-FUCCI*, *tubGAL80$^{ts20}$*. Detailed genotype of the fly lines used for the current work is listed in Key Resources Table.

Following genotypes were recombined for the current study:

1. *Antp-Gal4.UAS-mCD8-RFP/Tb*
2. *TRE-GFP/TRE-GFP; Antp-Gal4.UAS-mCD8-RFP/Tb*
3. *UAS-bsk DN/UAS-bsk DN; +/+; UAS-Relish RNAi/UAS-Relish RNAi*
4. *UAS-GMA/UAS-GMA; tubgal80ts/ tubgal80ts; Antp-Gal4 /Tb*
5. *w; pcol85-Gal4/UAS-2XeGFP; tub-Gal 80ts*
6. *UAS-Relish RNAi$^{KK}$/UAS-Relish RNAi$^{KK}$; UAS-Wg RNAi/ UAS-Wg RNAi*
7. *tubgal80ts/ tubgal80ts; Antp-Gal4.UAS-2XeGFP/TM2*
8. *UAS-Relish /UAS-Relish; UAS-EcR-DN/ UAS-ECR-DN.*
9. *UAS-ena RNAi$^{KK}$/cyo; UAS-Relish RNAi/Tb*
10. UAS-hep$^{act}$/FM7RFP;+/+; UAS-wg RNAi/Tb

All stocks were maintained at 25°C on standard media. For *GAL80$^{ts}$* experiments, crosses were initially maintained at 18°C (permissive temperature) for 2 days AEL to surpass the embryonic development and then shifted to 29°C till dissection.

For time series experiments, synchronization of larvae was done. Flies were allowed to lay eggs for about 4 hr. Newly hatched larvae within 1 hr intervals were collected and transferred onto food plates and kept at 29°C till dissection.

## Immunohistochemistry

Immunostaining and dissection (unless said otherwise) were performed using protocols described in *Jung et al., 2005*; *Mandal et al., 2007*; *Mondal et al., 2011* using primary antibodies: mouse anti-c-Rel (1:50, a gift from N.Silverman; *Stöven et al., 2000*), mouse anti Relish (1:50, 21F3, DSHB), mouse anti-Antp (1:10, 8C11, DSHB), mouse anti-Wg (1:3, 4D4, DSHB), mouse anti-P1 (1:40, a gift from I. Ando), rabbit anti-Ance (1:500, a gift from A. D. Shirras; *Hurst et al., 2003*), rat anti-Ci (1:5, 2A1, DSHB), mouse anti-singed (1:20, Sn7C, DSHB), mouse anti-enabled (1:30, 5G2, DSHB), rabbit anti-PH3 (1:150, Cell Signaling), rabbit anti-Hh (1:500, a gift from P. Ingham; *Forbes et al., 1993*), mouse anti-Hindsight (1:5, 1G9, DSHB), mouse anti-EcR common (1:20, DDA2.7, DSHB), mouse anti-β-PS (1:3, CF.6G11, DSHB), rabbit-anti-GFP(1:100, 2555, Cell Signalling), and rat anti-shg (1:50, DCAD2, DSHB). Secondary antibodies used in this study are as follows: mouse Cy3, mouse FITC, mouse Dylight 649, rabbit Cy3, (1:500), and rabbit-FITC (1:200) (Jackson Immuno-research Laboratories).

Tissues were mounted in Vectashield (Vector Laboratories) and then followed by confocal microscopy (LSM, 780, FV10i, LSM 900).

## EdU incorporation assay

Click-iT EdU (5-ethynyl-2'-deoxyuridine, a thymidine analog) kit from Life Technologies was used to perform DNA replication assay (*Milton et al., 2014*). Larval tissue was quickly pulled out in 1× PBS on ice (dissection time not more than 25 min and fat body and salivary gland needs to be cleared from the tissue of interest). Incubation of the dissected tissue was done in EdU solution, Component A (1:1000) in 1× PBS on shaker at room temperature for 30–35 min followed by fixation in 4% paraformaldehyde (prepared in 1× PBS). Post-fixation tissues were washed with 0.3% PBS-Triton four times at 10 min interval followed by 30–35 min of blocking in 10% NGS in 0.3% PBS-Triton. EdU staining solution as per manufacturer's instruction (for 50 µl staining solution, 43 µl 1× EdU buffer, 2 µl CuSO$_4$ solution, 5 µl 1× EdU buffer additive, 0.12 µl Alexa solution) was used to stain the sample for 30 min at room temperature. Two quick washes with 0.3% PBS-Triton was followed by a quick

wash in 1× PBS. If no further antibody staining was required, nuclear staining by DAPI was done in 1× PBS and then mounted in Vectashield.

## Extracellular Hh staining and quantitation

For extracellular Hh staining, a *detergent*-free staining protocol was used. Lymph glands were dissected in ice-cold Schneider's media (Gibco 21720024), rinsed with cold PBS twice, and fixed with 4% formaldehyde overnight at 4°C (*Sharma et al., 2019*). Subsequent processing of the samples was the same as mentioned above in the Immunohistochemistry section, except that no detergent was used.

Protocol described by *Ayers et al., 2010* was used as a reference to perform quantitation. A rectangle (500 × 150 pixels) was drawn, spanning from the niche to the cortical zone diagonally with the medullary zone in the middle, as shown in *Figure 4F*. An extracellular Hedgehog profile was made using the 'Plot Profile' tool of ImageJ. The Plot profile tool displays a 'column average plot', wherein the x-axis represents the horizontal distance through the selection and the y-axis the vertically averaged pixel intensity, which in this analysis is formed by extracellular Hedgehog staining.

## Filopodial detection and quantitation

*UAS-GMA* was used to label the filopodia using a niche-specific driver, *Antp-GAL4*. Lymph glands of the desired genotype were dissected in Schneider's media (Gibco 21720024) and incubated in a solution containing Schneider's media supplemented with 1% Phalloidin from *Amanita phalloides* (P2141 SIGMA) for 15 min in order to stabilize the filopodia. These tissues were then mounted and imaged directly under the confocal microscope.

The intact PSC cells are often scattered when we carry out a live analysis. This is mainly due to imaging requirements that demand a coverslip to be placed on the sample. The coverslip creates a pressure on the unfixed/live sample leading to the scattering of the cells.

Filopodial quantitation was done using ImageJ. The number of filopodia emanating from the niche in all the Z-stacks was counted manually per sample. The average number of filopodia emanating per sample was plotted using GraphPad Prism for different biological replicates. For filopodial lengths, the 'Freehand line' tool was used to mark the entire filopodial lengths, and the 'Measure' tool was employed to get values in μM. Filopodial lengths in all samples were then plotted collectively as individual points in GraphPad Prism.

## Phalloidin staining

Lymph glands dissected were fixed and incubated in rhodamine-phalloidin (1:100 in PBS) (Molecular Probes) for 1 hr. The samples were then washed thrice for 10 min in PBS followed by mounting in DAPI Vectashield before imaging.

## Quantification of intensity analysis of phalloidin

Membranous intensity of Phalloidin was measured using line function in Image J/Fiji. Mean intensity was taken in a similar manner as mentioned in *Shim et al., 2012*.

p-values of <0.05, <0.01, and <0.001, mentioned as *, **, and ***, respectively, are considered as statistically significant.

## Imaging and statistical analyses

All images were captured as Z sections in Zeiss LSM 780 confocal microscope and Olympus Fluoview FV10i (Panel 7). Same settings were used for each set of experiments. All the experiments were repeated at least thrice to ensure reproducibility. Mostly, 10 lymph glands were analyzed per genotype for quantification analysis. Data expressed as mean ± standard deviation of values from three sets of independent experiments. At least 10 images of the lymph gland/niche were analyzed per genotype, and statistical analyses performed employed two-tailed Student's t-test. p-values of <0.05, <0.01, and <0.001, mentioned as *, **, and ***, respectively, are considered as statistically significant. All quantitative analysis was plotted using GraphPad.

## Quantitative analysis of cell types in lymph gland

### PSC cell counting

Antp-positive cells were counted using the spot function in imaris software (*Sharma et al., 2019*). Data from three independent experiments are plotted in GraphPad prism as mean ± standard deviation of the values. All statistical analyses performed employing two-tailed Student's t-test. (http://www.bitplane.com/download/manuals/QuickStartTutorials5_7_0.pdf).

### Quantification of intensity analysis

Intensity analysis of Hh, TRE-GFP, Wg, Singed, Enabled, Relish antibody, and *Rel* transcript in different genotypes was done using protocol mentioned in http://sciencetechblog.files.wordpress.com/2011/05/measuring-cell-fluorescence-using-imagej.pdf. For each genotype, in about 10 biological samples, at least five ROIs were quantified. Data is expressed as mean ± standard deviation of values and are plotted in GraphPad prism. All statistical analyses performed employing a two-tailed Student's t-test.

### Differentiation index calculation

To calculate the differentiation index, middle most stacks from confocal *Z* sections were merged into a single stack for each lymph gland lobe using ImageJ/Fiji (NIH) software as described earlier (*Shim et al., 2013*). P1-positive area was marked by using Free hand tool. The size was measured using the Measure tool (Analyse–Measure). In similar way, DAPI area was also measured. The differentiation index was estimated by dividing the size of the P1-positive area by the total size of the lobe (DAPI area). For each genotype, mostly 10 lymph gland lobes were used, and statistical analysis was performed using two-tailed Student's t-test.

### *Fucci* cell cycle analysis

$UAS\text{-}GFP\text{-}E2f1_{1\text{-}230}$ $UAS\text{-}mRFP1NLS\text{-}CycB_{1\text{-}266}$ (*Zielke and Edgar, 2015*) fly line depends on GFP- and RFP-tagged degrons from E2F1 and Cyclin B proteins. Both E2F1 and Cyclin B gets degraded by APC/C and $CRL4^{cdt2}$ ubiquitin E3 ligases once they enter S and G2-M phase of cell cycle, respectively. Due to accumulation of $GFP\text{-}E2f1_{1\text{-}230}$, G1 phase will show green fluorescence, and due to accumulation of $mRFP1NLS\text{-}CycB_{1\text{-}266}$, S phase will show red fluorescence. Since both $GFP\text{-}E2f1_{1\text{-}230}$ and $mRFP1NLS\text{-}CycB_{1\text{-}266}$ are present in G2 and M phases, the cells will show yellow fluorescence. $UAS\text{-}GFP\text{-}E2f1_{1\text{-}230}$ $UAS\text{-}mRFP1NLS\text{-}CycB_{1\text{-}266}$ fly stock was recombined with *Antp-Gal4* and was crossed to *UAS-Relish RNAi* and $w^{1118}$ to ascertain the cell cycle status niche cells. All flies were kept at 25°C, and larvae were dissected 96 hr AEH.

### Generation of axenic batches

Germ-free batches were generated following the ethanol-based dechorination method provided in *Elgart et al., 2016*. According to this method, embryos were collected and washed using autoclaved distilled water to get rid of residual food particles. Embryos were further dechorinated for 2–3 min in 4% sodium hypochlorite solution. Once this is done, embryos were washed with autoclaved distilled water and were transferred to the sterile hood. Further manipulations were done inside the hood in order to avoid cross-contamination. Embryos were further washed twice with sterile water and were transferred into standard cornmeal food supplemented with tetracycline (50 µg/ml).

### Bacterial plating experiment

For plating experiments, three to five late third-instar larvae were washed in 70% ethanol twice for 2 min. Furthermore, the larvae were washed using sterile $H_2O$ twice for 2 min. After this surface sterilization, the larvae were transferred into LB media and were crushed thoroughly using a pestle. Once crushed, the homogenates were spread on LB agar media and was incubated for 3–4 days at 25°C.

### Measuring of bacterial content by qPCR

To measure bacterial composition in the gut, 12–15 third-instar larval guts were dissected and pooled, and DNA was isolated manually using the protocol provided by VDRC (https://stockcenter.vdrc.at/images/downloads/GoodQualityGenomicDNA.pdf) and followed by PCR analysis using species-specific primers. *Drosophila* actin was used as a control.

| S. no. | Gene/species name | Primer sequence |
|---|---|---|
| 1 | Actin | 5'-GGAAACCACGCAAATTCTCAGT-3'<br>5'-CGACAACCAGAGCAGCAACTT-3' |
| 2 | Acetobacter | 5'-TAGTGGCGGACGGGTGAGTA-3'<br>5'-AATCAAACGCAGGCTCCTCC-3' |
| 3 | Lactobacillus | 5'-AGGTAACGGCTCACCATGGC-3'<br>5'-ATTCCCTACTGCTGCCTCCC-3' |

## Infection experiments

The following bacterial strains were used for infection: *E. coli* ($OD_{600}$:100). For larval infection, bacterial cultures were concentrated by centrifugation; the pellet formed was resuspended in phosphate-buffered saline (PBS) to appropriate OD value. Synchronized third-instar larval batches were used for all analyses. Third-instar larvae were washed three times with sterile $ddH_2O$ and pricked using a fine insect pin dipped in bacterial suspension at the postero-lateral part. Mock injections were done using PBS dipped pins. Complete penetration was confirmed while dissection by looking at the melanization spots at the larval epithelial surface. Once infected, larval batched were transferred to food plates and incubated at 25°Celsius till dissection. All observations were made 4 hr post-infection.

## IF-fluorescence in situ hybridization

The protocol we followed was modified from *Toledano et al., 2012*.

## Probe preparation

Rel clone was procured from DGRC. Following plasmid linearization and restriction digestion using EcoRV and Xho1, the DNA fragments were loaded in agarose gel for electrophoresis. Furthermore, the desired DNA fragments were purified using PCI (phenol:chloroform:isoamyl alcohol) based gel purification and DIG-labeled RNA anti-sense, and sense probe was prepared using Sp6 and T7 polymerase enzyme, respectively. Following DNase treatment, the probes were precipitated using LiCl and ethanol. The RNA pellet was dried resuspended in RNase-free $dH_20$ and stored at $-80°C$ till further use.

## Dual labeling of mRNA and protein in the hematopoietic niche

For IF-FISH, we followed the Part B of the Tissue preparation and fixation section of *Toledano et al., 2012*. Followed by quick dissection, the larval tissues (make sure of having minimum fat body cells since it can hinder the fixation and hybridization) were fixed for 30 min in 4% formaldehyde prepared in RNase-free PBS and further washed in PBTH (PBS containing 0.1% Tween 20 and 250 µg/ml yeast tRNA) for thrice, 10 min each. Samples were blocked using 5% BSA prepared in PBTHR (PBTH containing 0.2 U ml$^{-1}$ RNase inhibitor and 1 mM DTT). Furthermore, tissues were incubated in rabbit anti-GFP (1:100, prepared in PBTHR) for 18 hr at 4°C. Tissues were washed using PBTH three times 10 min each, followed by blocking for 30 min using 5% BSA prepared in PBTHR. The tissues were then incubated in fluorescent-labeled secondary antibody (rabbit-FITC 1:100) for 4 hr at room temperature in a shaker. Following this, three washes of PBTH, 10 min each, tissues were fixed using 10% formaldehyde for 30 min. Post-fixation, tissues were washed thrice, 5 min each and rinsed with 0.5 ml of prewarmed hybridization buffer (HB) for 10 min in a 65°C in a hybridization chamber. Tissued were then blocked with PHB (HB mixed with tRNA [10 mg/ml]) for 1 hr in 65°C. Following blocking, tissues were transferred to preheated RNA probe prepared in PHB (2 µg/ml) and incubated at 65°C for 18 hr. Post-hybridization, stringent washes were given using 0.1% PBT: HB mix as mentioned in *Toledano et al., 2012*. The issues were then blocked in TNB buffer for 1 hr prior to incubation anti-DIG-POD (1:1000) for 18 hr at 4°C. Post-primary antibody incubation, tissues were washed using 0.1% PBT. For signal detection and amplification, Alexa Fluor 594 Tyramide Reagent was used. Tyramide amplification solution was prepared as mentioned in the user guide. Tissues were incubated in TSA working solution for 8 min. Following this, an equal amount of Reaction stop

reagent solution was added and further incubated for 1 min. Post-TSA reaction, tissues were PBS rinsed thrice for 5 min and mounted in Vectashield.

## Acknowledgements

We thank B Lemaitre, N Silverman, I Ando, U Banerjee, S Cohen, P Ingham, M Crozatier, A D Shirras, R Schulz and NG Prasad for reagents. Special thanks to Sushmit Ghosh and Kaustuv Ghosh for assisting with FISH and germ-free experiments. We thank all members of the two laboratories for their valuable inputs. We thank IISER Mohali's Confocal Facility, Bloomington *Drosophila* Stock Center, at Indiana University, VDRC (Vienna) and Developmental Studies Hybridoma Bank, University of Iowa for flies and antibodies. Models 'Created with BioRender.com'. DBT Wellcome-Trust India Alliance Senior Fellowship [IA/S/17/1/503100] to LM and Institutional support to PR, NSD, SM, and DBT-fellowship funding to AK for this study duly acknowledged.

## Additional information

### Funding

| Funder | Grant reference number | Author |
| --- | --- | --- |
| The Wellcome Trust DBT India Alliance | IA/S/17/503100 | Lolitika Mandal |

The funders had no role in study design, data collection and interpretation, or the decision to submit the work for publication.

### Author contributions

Parvathy Ramesh, Conceptualization, Data curation, Formal analysis, Validation, Investigation, Visualization, Methodology, Writing - original draft, Writing - review and editing; Nidhi Sharma Dey, Formal analysis, Validation, Investigation, Visualization, Methodology; Aditya Kanwal, Conceptualization, Validation, Investigation, Visualization, Methodology, Writing - original draft; Sudip Mandal, Formal analysis, Writing - original draft, Writing - review and editing; Lolitika Mandal, Conceptualization, Data curation, Formal analysis, Supervision, Funding acquisition, Validation, Investigation, Visualization, Methodology, Writing - original draft, Project administration, Writing - review and editing

### Author ORCIDs

Parvathy Ramesh https://orcid.org/0000-0001-7273-5848
Nidhi Sharma Dey http://orcid.org/0000-0001-9432-0221
Aditya Kanwal https://orcid.org/0000-0002-8066-6798
Sudip Mandal http://orcid.org/0000-0002-2211-483X
Lolitika Mandal https://orcid.org/0000-0002-7711-6090

### Decision letter and Author response

Decision letter https://doi.org/10.7554/eLife.67158.sa1
Author response https://doi.org/10.7554/eLife.67158.sa2

## Additional files

### Supplementary files

• Transparent reporting form

### Data availability

All data generated or analysed during this study are included in the manuscript and supporting files. Source data files have been provided for Figures 1- 7 and their respective figure supplements.

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
