## [Decision Letter]

**Acceptance summary:**

Relish is one of the three NK–kB factors in *Drosophila* of which functions are well–known in the insect immunity. This study identified how Relish in the *Drosophila* larval hematopoietic niche contributes to hematopoiesis during normal growing conditions and how it achieves a switch to immune responses upon bacterial infections. The primary claims are well–supported, and this study will highly contribute to insect hematopoiesis and immunity.

**Decision letter after peer review:**

[Editors’ note: the authors submitted for reconsideration following the decision after peer review. What follows is the decision letter after the first round of review.]

Thank you for submitting your work entitled "The dynamic role of Relish in the niche to modulate *Drosophila* blood progenitor homeostasis in development and infection" for consideration by *eLife*. Your article has been reviewed by 2 peer reviewers, and the evaluation has been overseen by a Reviewing Editor and a Senior Editor. The reviewers have opted to remain anonymous.

Our decision has been reached after consultation between the reviewers. Based on these discussions and the individual reviews below, we regret to inform you that your work will not be considered further for publication in *eLife*. Nevetherless, you still has the possibility to re–submit a revised version but this will be considered as a new submission.

The two reviewers were quite enthusiast about the manuscript but they also find that there are too many issues to solve before acceptation that cannot be addressed in the two month time frame. Note that you still have the opportunity to re–submit a revised manuscript to *eLife* that will be considered as a new submission and likely be reviewed by the same reviewers. Unfortunately, we did not get answer from a third reviewer. In the eventually of a resubmission, we will contact an additional reviewer. At this stage you should decide if you wish to submit a revised manuscript that address the point or move your paper to another journal.

Essential revisions:

*Reviewer #1:*

In this manuscript, Mandal and colleagues identified a novel role for Relish in the hematopoietic niche development and its coordinative function with immune responses. The authors found that Relish is enriched in the PSC and loss of which causes a dramatic reduction of hematopoietic progenitors. Interestingly, the loss of Relish in the PSC causes the proliferation of PSC cells through activation of Wg and alters the actin cytoskeletal structure of the PSC, in turn trapping Hh in the PSC. The authors also observed that these phenotypes are led by ectopic activation of *JNK* through *TAK1*. The authors moved on to find the developmental and physiological relevance of this phenomenon and discovered that EcR signaling activates Rel in the PSC during normal development while innate immunity attenuates the Rel expression in the PSC.

This study first describes the novel role of Relish in blood development and its association with ecdysone signaling as well as with innate immunity. The study is interesting, well–designed and the mechanism shown is novel enough to merit the journal. But the last part, where the biological significance of Rel in the PSC is described, is rather weak compared to the other genetic interactions and require further assessments or better description.

1. Although the authors focus on the expression of Relish in the PSC, Relish is also expressed in progenitors, as shown in Figures 1C–D and Figure 2–supplement 1A–E. Related, loss of Rel in the PSC seems to attenuate Rel in both the medullary zone and the PSC (Figure 1D). Is the Relish expression coordinated in both zones? Does Rel RNAi lead to similar phenotypes when driven in the progenitors? Is the Rel function shown cell–autonomous?

2. Compared to the genetic assessments, paragraphs describing the developmental and immunological relevance of Relish are rather weak and overstated. The only genetic basis shown for the Relish and ecdysone interaction is the expression of UAS–EcR_DN. Does UAS–relish rescue the EcR_DN phenotypes? If indeed ecdysone modulates the Relish level, is there any oscillation in the level of Relish in the PSC during larval development? How does Rel–*TAK1*–*JNK*–Wg or Hh axis change according to the ecdysone level during normal development? Are Relish and its downstream pathway up or downregulated in the PSC when ecdysone is additionally given?

3. Related to #2, the authors proposed that proteasomal degradation (Factor X) will downregulate Relish during bacterial infection. Have the authors verified that the other components in the pathway upon infection? Or is it already known? Is overexpression of UAS–Relish_WT sufficient to block the precocious differentiation of lymph gland cells upon infection? Does ecdysone level change during infection? Without showing direct evidence, lines 475–489 should be moved to the discussion, and the results need to be toned down.

*Reviewer #2:*

In this manuscript the authors investigate the role of Relish in the *Drosophila* lymph gland (LG). They establish that relish is expressed in PSC cells and that reducing its expression in these cells (by expressing relish RNAi with a PSC–gal4 driver) leads to an enlarged PSC, increased plasmatocyte differentiation, no effect on crystal cell numbers, and fewer progenitors in the medullary zone (MZ). In the PSC, Relish controls Wingless levels that in turn control PSC cell proliferation and thus PSC size. This study also establishes that the knock down of relish in the PSC leads to increased levels of several actin binding proteins, reduced filopodia formation in PSC cells and a decrease in Hh (HhExt) release from the PSC. In addition, relish knock–down in the PSC leads to the activation of the *JNK* pathway in the PSC. Epistasis experiments establish that *JNK* acts downstream of Relish to control filopodia formation and HhExt. Under normal conditions, Relish levels in the PSC are under the control of ecdysone. Finally, in response to an *E. coli* infection, a decrease in Relish levels in the PSC is observed together with increased plasmatocyte differentiation.

This is an important study describing a yet unknown regulation of *Drosophila* LG hematopoiesis. However, I have some concerns with the current version of the manuscript.

In the introduction: the authors do not state that the role of the PSC in the LG under normal conditions is under debate, and papers relative to this, even if they diverge from the dogma, must be cited. It is established in the literature that the MZ progenitors are heterogeneous; this information together with the corresponding papers should be introduced. In the Results section, my first concern is about the models proposed in Figure 7: many epistasis experiments are lacking, and thus at this stage of the analysis it is impossible to propose such a model (see comments below). Furthermore, several controls (pictures and quantifications) are lacking (see below).

The second problem concerns the relationships between relish knock–down in the PSC, the absence of filopodia and the decrease of HhExt. These are important and novel data and should be given in the main figures. However, they need to be consolidated (see comments below).

The third point relates to the role played by Relish in the PSC in response to bacterial infection. In the current version of the manuscript, the data presented are too preliminary to prove that Relish is required and to propose how it is involved in the control of LG stress hematopoiesis.

Figure 1: In the PSC>rel RNAi context, the consequence on blood progenitors must be analyzed in more detail Indeed, it is possible that an increase in the mature plasmatocytes numbers results from increased CZ (cortical zone) precursor maturation, without an input from the MZ pool. A systematic analysis of MZ markers (and quantifications) should be performed in all the genetic contexts analyzed (see below for other figures). Additional markers for MZ cells, other than Ci, must be tested since Ci is part of the Hh signaling pathway which is impaired in this context. Why is the crystal cell index unchanged in the PSC >rel RNAi context? Is there lamellocyte differentiation?

Figure 3 : What about the contribution of PSC Wg in the control of LG homeostasis? What about LG homeostasis (MZ and differentiated blood cells) under conditions where the PSC size is rescued (i.e.: PSC>rel RNAi, Wgts)? In Figure 3 D–G, the control Wgts (picture and quantification) is missing.

Figure 3B–B': Why is Wg staining stronger in PSC cells that express lower levels of GFP?

Figure 3 sup1 E: Is there a significant difference in PSC cell numbers between antp>wgRNAi and control LGs? This should be indicated. In Figure 3 sup 1 F–I: What about crystal cell differentiation and MZ progenitors in PSC>wgRNAi and PSC>WgRNAi, relRNAi?

The control i.e. PSC>wg RNAi (both pictures and quantifications) is missing. Is there any difference in LG size in this context as previously reported by Sinenko et al., 09?

Figure 4 : Are there LG defects when the levels of actin binding proteins (F actin, singed,..) are modified in the PSC ? Is there a functional relationship between Relish and actin binding protein levels in the PSC?

Figure 4 A–B : Is Hh expressed in a subset of CZ cells? Is there basal expression of Hh also in the MZ (see A' and B")?

Figure4 sup1 A–E: Data relative to filopodia formation and HhExt must to be given in the main figures since they are important results. However, they still need to be consolidated. Quantifications for HhExt (quantity, dispersion) and filopodia (numbers and size) are necessary.

Pictures A–A": Is HhExt, as shown with a comet–like pattern outside the PSC, always observed in controls, or is it specific to this picture?

Pictures C–C': These is a problem here: the cells that are shown become detached from the PSC and migrate. In such conditions there are more cytoplasmic extensions that reflect the migrating status of the cells. Therefore this is not a correct illustration for PSC cell filopodia.

Pictures F–F' '(as well as in Figure 4G–L): There are discrepancies between pictures and quantifications (a 6 fold increase in the PSC>rel RNAi context compared to the control is not observed in the pictures).

Figure 5 H–L: the control, corresponding to PSC> bskDN (pictures and quantifications) is missing. Figure F5 O–Q, the control, *tak1*/+ (pictures and quantifications) is missing.

Figure 5 L–L', should be replaced (blurry picture and migrating PSC cells).

What about MZ and crystal cell differentiation in these different conditions ( i.e: UAS–bskDN, UAS bskDN, Rel RNAi; *tak1*/+ and *tak1* , rel RNAi)?

–Epistasis between the *JNK* pathway and Hh, and the *JNK* pathway and wg must be analysed. This is essential to support the models given in Figure 7.

Figure 6: The demonstration that Ecdysone controls relish transcription in the PSC is lacking Analysis of Relish protein levels is not enough, since post transitional regulation has been described for Relish. relish transcription must be analyzed in PSC>EcRDN LGs. Furthermore, epistasis experiments are required to firmly establish a functional link between Relish and ecdysone.

Figure 7: The link between Relish in the PSC and its role in LG hematopoiesis in response to bacterial infection is not yet clearly assessed. That an increase in blood cell differentiation occurs in the LG in response to bacterial infection has already been described (Khadilkar et al;, 2017). This reference should be cited in the manuscript. Whether the bacteria–induced decrease of Relish in the PSC is involved in the LG stress hematopoiesis is not established here since there are many discrepancies between LG phenotypes in PSC>rel RNAi larvae and in response to bacterial infection and rescue experiments are lacking. While in both cases there is an increase in plasmatocyte differentiation, there is no difference in PSC size in response to bacterial infection (which differs from PSC>rel RNAi LGs), and there is no LG disruption in PSC>rel RNAi larvae (which differs from bacterial infection). Furthermore, the LG disruption in response to bacterial infection is a novel data, but unfortunately it is not properly documented in this manuscript. Finally, in bacterial infection experiments, no data relative to a role of Hh or *JNK* pathway in PSC cells have ever been described so far.

Is the transcription of relish or the stability of the protein that is affected in response to bacterial infection? In Figure 7 D: >UAS –Rel 68KD, without infection, is Relish protein detected by immunostaining in PSC cells? This would clarify whether Relish protein stability is dependent on bacterial infection.

Figure 7 B–D: Both quantifications of PSC cell numbers and Relish levels are missing.

Figure 7 H–J': Plasmatocyte quantification is missing. Analyses and quantifications of crystal cells and MZ progenitors must also be provided.

Models in Figure A and K are not correct, since many epistasis experiments are missing.

Furthermore, an arrow between Wg in the PSC and MZ progenitors is missing, since it has been previously established that the Wg pathway in the PSC controls LG hemocyte differentiation (please see Sinenko et al., 2009).

Figure 8 : Are there any functional links between filopodia, the actin skeleton and Hh signaling in the LG ? This point should be further addressed.

For the PSC>rel RNAi context, both in the summary and in point e in the legend Figure 8, we read "precocious differentiation of the blood progenitors". This point should be better established. Indeed, systematic analyses of MZ markers in the different genetic contexts are required (see comments above).

[Editors’ note: further revisions were suggested prior to acceptance, as described below.]

Thank you for submitting your article "Relish plays a dynamic role in the niche to modulate *Drosophila* blood progenitor homeostasis in development and infection" for consideration by *eLife*. Your article has been reviewed by 2 peer reviewers, and the evaluation has been overseen by a Reviewing Editor and Utpal Banerjee as the Senior Editor. The reviewers have opted to remain anonymous.

Essential revisions:

General comments by the editor following the discussion with the reviewers

Both reviewers feel that this paper warrants publication in *eLife*.However , they think that additional new data are needed to fully support the conclusion on the role of PSC fillopodia in signaling. At the very least, the authors must establish whether there is a functional relationship between Relish and actin binding protein levels in the PSC. Rescue experiments of relish knock down, by decreasing "actin protein" and looking at lymph gland homeostasis, fillopodia formation and Hh diffusion, have to be provided.

No functional relationship has been established for the role of Relish in the PSC in response to bacterial infection Thus, the authors should be more cautious, and both in the manuscript and the summary the text relative to this point has to be modulated. Furthermore, the authors should discuss the discrepancy between their model, where in response to bacterial infection Hh is trapped within the niche and unable to diffuse, versus the model proposed by (Khadilkar et al., 2017) where bacterial infection causes the breakdown of the permeability barrier in PSC cells.

To minimize the second round of revision, the reviewers suggested that the authors could also tone down the statements on the filopodia–Hh interaction and the role of Rel in the bacterial infection while adding possible additional works.

1) As I already suggested in the previous comment, the authors should impair PSC fillopodia formation by modifying actin binding proteins (F actin, singed,dia,…) and analysing both lymph gland defects and Hh diffusion. It is also necessary to establish whether there is a functional relationship between Relish and actin binding protein levels in the PSC. Rescue experiments of relish knock down, by decreasing "actin protein" and looking at lymph gland defect, fillopodia formation and Hh diffusion are missing.

In Figure 4 Sup1: In PSC>UAS Dia RNAi, what about fillopodia formation and Hh diffusion ? Does decreasing dia in PSC cells in relish knocked down condition restore fillopodia formation, Hh diffusion and lymph gland cell homeostasis? This is an important issue than remains to be addressed.

Figure 4A–C: when relish is knocked down in the PSC, is Hh the increase in PSC cells due to a defect of Hh diffusion and/or a decrease in Hh transcription in those cells? This has to be clarified.

2) The role of Relish in the PSC in response to bacterial infection is based on correlation, since rescue experiments are missing and thus no functional relationship has been established yet. Furthermore, whereas bacterial infection leads to the increase in crystal cell differentiation (Khadilkar et al., 2017), relish knock down in PSC cells has no impact on crystal cell number. Thus, the authors should be more cautious, and both in the manuscript and the summary the text relative to this point has to be modulated. Finally, the authors should discuss the discrepancy between their model where in response to bacterial infection Hh is trapped within the niche and unable to diffuse, versus the model proposed by (Khadilkar et al., 2017) where the bacterial infection causes the breakdown of the permeability barrier in PSC cells.

---

## [Author Response]

[Editors’ note: the authors resubmitted a revised version of the paper for consideration. What follows is the authors’ response to the first round of review.]

Essential revisions:Reviewer #1:This study first describes the novel role of Relish in blood development and its association with ecdysone signaling as well as with innate immunity. The study is interesting, well–designed and the mechanism shown is novel enough to merit the journal.

Thank you very much for your comments. We are happy to note that our effort is well appreciated.

But the last part, where the biological significance of Rel in the PSC is described, is rather weak compared to the other genetic interactions and require further assessments or better description.

Thanks for your input. We have addressed all the concerns related to the biological significance of Rel in the hematopoietic niche/PSC. All the experiments suggested from your end have been performed to add strength to this part of our manuscript. Thanks a lot for enriching our study.

1. Although the authors focus on the expression of Relish in the PSC, Relish is also expressed in progenitors, as shown in Figures 1C–D and Figure 2–supplement 1A–E. Related, loss of Rel in the PSC seems to attenuate Rel in both the medullary zone and the PSC (Figure 1D). Is the Relish expression coordinated in both zones? Does Rel RNAi lead to similar phenotypes when driven in the progenitors? Is the Rel function shown cell–autonomous?

a. Loss of Relish expression in progenitor cells of MZ is due to the drastic decline in their number. Therefore the loss of Rel expression in the MZ is not per cell instead due to fewer progenitor cells.

b and d. To check whether the expression and function of Relish in the niche and Medullary zone are mutually independent, we checked the intensity of Relish protein expression in the niches and progenitors of both control and Rel RNAi lymph glands. Quantitative analysis revealed more than a 2-fold reduction in Relish expression in the Rel RNAi niches compared to control, whereas progenitor specific expression remained unaltered (please refer to Figure 1 C-E). The above observation itself indicates that the Rel function described in our study is cell-autonomous.

c. The Rel function within the progenitors is entirely independent to one of the niche. It is a part of an ongoing investigation in the laboratory and is beyond the scope of the current manuscript.

2. Compared to the genetic assessments, paragraphs describing the developmental and immunological relevance of Relish are rather weak and overstated. The only genetic basis shown for the Relish and ecdysone interaction is the expression of UAS–EcR_DN. Does UAS–relish rescue the EcR_DN phenotypes? If indeed ecdysone modulates the Relish level, is there any oscillation in the level of Relish in the PSC during larval development?

Thanks for raising this issue. In the current version of the manuscript, these missing pieces of information have been included.

Upon niche-specific overexpression of Relish in conjunction with EcR loss, we found a rescue in niche proliferation (Figure 6 Q-U) and differentiation defects (Figure 6 figure supplement 2A-E), which are otherwise associated with EcR loss (Figure 6 I-L and M-P). These are also mentioned in Main text Line: 407-413.

As far as Rel expression is concerned, we assayed the Rel protein at two different timepoints that 60 and 72hrs AEH which corresponds to Ecdysone crest and trough respectively. We do not see any significant difference in the level reflecting oscillation. We think that the basal level of Ecdysone present throughout the larval stages (Hodgetts et al., 1976; Kraminsky et al., 1980, Handler, 1982 Riddiford, L. M., 1994) is sufficient to sustain Rel expression in the niche. Please refer to Author response image1A-C below.

**Author response image 1. sa2fig1:** A-C: No significant change in Relish expression in the niche was observed at 72 hours (AEH) (B-B') compared to 60 hours (AEH) ( A-A'). Statistical analysis of the data from A-B' (n=29 P-value =.297 ; two tailed Students t-test).

How does Rel–TAK1–JNK–Wg or Hh axis change according to the ecdysone level during normal development? Are Relish and its downstream pathway up or downregulated in the PSC when ecdysone is additionally given?

Our study reveals that during development, ecdysone-mediated Relish expression represses *JNK* activity in the niche. Therefore, in the absence of Rel, there is an ectopic activation of *JNK* signaling (Figure 5A-C). Thus, it can be speculated that an exogenous supply of ecdysone will increase the Relish level in the niche, strengthening the inhibitory arm on *JNK* signaling. Based on the existing literature (Karim et al.,1991, A.J Andres A.J and Thummel C S, 1994 ), we incubated lymph glands in 20E (5uM). A slight increase in Rel expression was consistently seen compared to mock treatment in these ex vivo experiments (See below, Figure 2 A-D). In sync with our result, *JNK* activation within the niche was not seen even in this scenario. This observation further

endorsed our claim that Rel expression can prevent *JNK* activation (See Author response image 2).

**Author response image 2. sa2fig2:** A-C': Post 20E incubation, slight increase in Relish expression was observed compared (B-C') to mock incubated samples (A-A'). D. Statistical analysis of the data from A-B' (n=24 P-value=6.19 x10-7; two tailed Students t-test). E-F. *JNK* expression remained unaltered in 20E incubated (F) and mock (E) incubated samples.

3. Related to #2, the authors proposed that proteasomal degradation (Factor X) will downregulate Relish during bacterial infection. Have the authors verified that the other components in the pathway upon infection? Or is it already known? Is overexpression of UAS–Relish_WT sufficient to block the precocious differentiation of lymph gland cells upon infection? Does ecdysone level change during infection? Without showing direct evidence, lines 475–489 should be moved to the discussion, and the results need to be toned down.

In the uninfected scenario, overexpressing Relish in the niche decreased niche cell number and progenitor differentiation (Figure 1 figure supplement 1F-H and Figure 6 figure supplement 2C and E). However, during infection, Relish gets degraded even in overexpression scenario compared to sham (Figure 7H-H'). Therefore, we can conclude that post-transcriptional regulation is affecting Relish stability during infection.

Based on EcR common expression (validated reporter for Ecdysone activity, Hogness DS et al.,1993, Segraves WA et al.,1999, Matunis EL et al.,2014), it is evident that the level of Ecdysone signaling remains unaltered during infection. However, subsequent loss of Relish expression points out to a regulation evoked during infection to override the developmental input (See Author response image 3).

**Author response image 3. sa2fig3:** A-B: No significant change in EcR expression in the niche was observed upon infection and sham. C. Statistical analysis of the data from A-B (n= 12 P-value=.364).

Reviewer #2:In this manuscript the authors investigate the role of Relish in the Drosophila lymph gland (LG). They establish that relish is expressed in PSC cells and that reducing its expression in these cells (by expressing relish RNAi with a PSC–gal4 driver) leads to an enlarged PSC, increased plasmatocyte differentiation, no effect on crystal cell numbers, and fewer progenitors in the medullary zone (MZ). In the PSC, Relish controls Wingless levels that in turn control PSC cell proliferation and thus PSC size. This study also establishes that the knock down of relish in the PSC leads to increased levels of several actin binding proteins, reduced filopodia formation in PSC cells and a decrease in Hh (HhExt) release from the PSC. In addition, relish knock–down in the PSC leads to the activation of the JNK pathway in the PSC. Epistasis experiments establish that JNK acts downstream of Relish to control filopodia formation and HhExt. Under normal conditions, Relish levels in the PSC are under the control of ecdysone. Finally, in response to an E. coli infection, a decrease in Relish levels in the PSC is observed together with increased plasmatocyte differentiation.This is an important study describing a yet unknown regulation of Drosophila LG hematopoiesis.

Thanks so much for your comments and appreciation. We are happy to note that we have been able to convey our findings to you.

However, I have some concerns with the current version of the manuscript.In the introduction: the authors do not state that the role of the PSC in the LG under normal conditions is under debate, and papers relative to this, even if they diverge from the dogma, must be cited. It is established in the literature that the MZ progenitors are heterogeneous; this information together with the corresponding papers should be introduced.

We have included the relevant references in the introduction section.

In the Results section, my first concern is about the models proposed in Figure 7: many epistasis experiments are lacking, and thus at this stage of the analysis it is impossible to propose such a model (see comments below).

Thanks for your input. Request you to kindly go through the current manuscript, which provides the necessary genetic and epistatic analysis to establish Ecdysone-Rel-*JNK* axis in the niche. (Line No: 401-413 and Figure 6Q-U, Figure 6 figure supplement 2A-E and Figure 6 figure supplement 2K-L'' ).

Furthermore, several controls (pictures and quantifications) are lacking (see below).

We want to mention that controls and their quantitation were done in all the cases. Due to lack of space, we did not include all the controls and their quantitation in the previous version. However, as per your suggestion in the revised version, all control and quantitation have been included in the main or Supplementary panels.

The second problem concerns the relationships between relish knock–down in the PSC, the absence of filopodia and the decrease of HhExt. These are important and novel data and should be given in the main figures. However, they need to be consolidated (see comments below).

Thanks for your input. We have now shifted these figures to the Main panel (details given below in). As advised, we have now included proper quantitation of Hh^Extra^ and that of the filopodia (numbers and length). We profusely thank the Reviewer for this input, as the outcome has further consolidated our claim.

The third point relates to the role played by Relish in the PSC in response to bacterial infection. In the current version of the manuscript, the data presented are too preliminary to prove that Relish is required and to propose how it is involved in the control of LG stress hematopoiesis.

Based on the input of both of the reviewers, we have strengthened this section of our manuscript. Kindly see the response below.

Figure 1: In the PSC>rel RNAi context, the consequence on blood progenitors must be analyzed in more detail Indeed, it is possible that an increase in the mature plasmatocytes numbers results from increased CZ (cortical zone) precursor maturation, without an input from the MZ pool. A systematic analysis of MZ markers (and quantifications) should be performed in all the genetic contexts analyzed (see below for other figures). Additional markers for MZ cells, other than Ci, must be tested since Ci is part of the Hh signaling pathway which is impaired in this context. Why is the crystal cell index unchanged in the PSC >rel RNAi context? Is there lamellocyte differentiation?

Regarding MZ markers, we have now included DE-Cad/Shotgun (Shg) immunostaining followed by the quantification of the same (Figure 1H-I' and J). Regarding crystal cell number, other than few outliers there is no significant change in the number. Lamellocyte differentiation was not observed in Rel loss scenario (βPS, Figure1 figure supplement 1I-J'). Thus, the ectopic differentiation seen upon Rel loss from niche is biased towards plasmatocyte fate.

Figure 3 : What about the contribution of PSC Wg in the control of LG homeostasis? What about LG homeostasis (MZ and differentiated blood cells) under conditions where the PSC size is rescued (i.e.: PSC>rel RNAi, Wgts)? In Figure 3 D–G, the control Wgts (picture and quantification) is missing.

a. Sinenko et al., 2010 has shown that downregulation of Wg signaling (by overexpression of Fz2DN) from the niche caused reduction in the progenitor pool. In sync with this finding, our result (employing UAS-wg-RNAi as well as a temperature-sensitive allele of wg) shows a decline in progenitor number (Shg, Figure 3I-M) and increased differentiation (P1, Figure 3 figure supplement 1B-F).

b. Curtailing wingless signaling in conjunction with the Rel downregulation from the niche rescues hyperproliferative niche but not the precocious differentiation of the progenitors (Figure 3 figure supplement 1G-J, L-P, and Q-U). Since maintenance of the progenitors is also dependent on Hh signaling from the niche, which is majorly affected upon Rel loss, is the reason for the failure in progenitor maintenance (Figure 3I-M and Figure 3 figure supplement 1L-P).

c. We have included the picture and the quantitation (Figure 3E-E' and H).

Figure 3B–B': Why is Wg staining stronger in PSC cells that express lower levels of GFP?

All niche cells do not express similar levels of Wingless protein.

Figure 3 sup1 E: Is there a significant difference in PSC cell numbers between antp>wgRNAi and control LGs? This should be indicated. In Figure 3 sup 1 F–I: What about crystal cell differentiation and MZ progenitors in PSC>wgRNAi and PSC>WgRNAi, relRNAi?

There was a significant decrease in niche cell proliferation in wg^ts^ (Figure 3E-E' and H), while AntpGal4>UASwgRNAi was almost comparable to control (Figure 3 figure supplement 1 I and K). However, upon co-expression with UAS-RelRNAi, the niche number was significantly restored (Figure 3G-G' and H). Interestingly, downregulating wg function in Rel loss genetic background could not rescue the ectopic differentiation of progenitors (Shg, Figure 3 figure supplement 1O and P, Figure 3 figure supplement 1T and U). The number of progenitors as reflected by Shg in this genotype is less compared to control.

Upon loss of wg from the niche, there was a slight decrease in crystal cell index; however, in conjunction with Rel loss from the niche, no significant difference with the control was observed (see Author response image 4).

**Author response image 4. sa2fig4:** A-D. Slight decrease in Crystal cell index was observed in wg-RNAi (C) compared to control (C) whereas no significant change was observed in Rel RNAi(KK) (B) and Rel RNAi(KK), wg RNAi (D). E. Statistical analysis of the data from A-D (n=11 P–value=3.4x10-2 for control versus wg-RNAi).

The control i.e. PSC>wg RNAi (both pictures and quantifications) is missing. Is there any difference in LG size in this context as previously reported by Sinenko et al., 09?

Loss of wg function through wg^ts^, we do find a significant decrease in the size of the lymph gland as compared to control (see Author response image 5). Please note that we have used a temperature sensitive allele of wg whereas Sinenko et al., had downregulated wg function by overexpressing Dfz2DN from the PSC.

**Author response image 5. sa2fig5:** Significant decrease in LG area was observed in wgts compared to control (n=10, P-value = 2. 3x10-4 two tailed students t-test).

Figure 4 : Are there LG defects when the levels of actin binding proteins (F actin, singed,..) are modified in the PSC ? Is there a functional relationship between Relish and actin binding protein levels in the PSC?

Upon niche-specific downregulation of Diaphanous (an actin polymerase), a significant increase in the differentiation compared to control (Figure 4 figure supplement 1A-B' and C) is observed.

Based on our genetic data, we would like to infer that the actin remodeling observed in Rel loss is *JNK* dependent.

Figure 4 A–B : Is Hh expressed in a subset of CZ cells? Is there basal expression of Hh also in the MZ (see A' and B")?

CZ specific expression of Hh maps to the subset of crystal cells since it colocalizes with Lozenge (a validated marker for crystal cell). The basal level is the extracellular Hh diffused from the source (niche) and is sensed by its receptor patched expressed in the MZ cells.

**Author response image 6. sa2fig6:** Crystal cells marked by Lz>GFP also expresses Hh (red).

Figure4 sup1 A–E: Data relative to filopodia formation and HhExt must to be given in the main figures since they are important results. However, they still need to be consolidated. Quantifications for HhExt (quantity, dispersion) and filopodia (numbers and size) are necessary.

Thanks you for your suggestions. We do understand your point. We have done the necessary inclusions. Please refer to Main text: Line No 254-271.

We have also provided the Intensity analysis of Hh^Ext^ Figure 4F, and quantitative analysis of filopodial number and length in the current version (Figure 4J-K).

Pictures A–A": Is HhExt, as shown with a comet–like pattern outside the PSC, always observed in controls, or is it specific to this picture?

Sorry for the confusion created. The comet-like pattern is generated due to the inclusion of the multiple stacks. We have now processed the few confocal stacks of the same image to get a better representation. Please look at the new figure (Figure 4 D-D'') in the revised manuscript. For your reference, the old and the new are shown below.

Pictures C–C': These is a problem here: the cells that are shown become detached from the PSC and migrate. In such conditions there are more cytoplasmic extensions that reflect the migrating status of the cells. Therefore this is not a correct illustration for PSC cell filopodia.

We want to draw the Reviewer's attention towards the fact that the imaging has been done on live tissues to preserve and document the dynamic and fragile filopodial extensions. As soon as we put the coverslip in live conditions, the pressure created tends to detach the niche cells from each other in the control lymph glands. However, the niche cells upon Relish loss remain attached to each other. This observation can be explained by our results, which demonstrate that loss of Rel from the niche cells increases cortical actin accumulation, forcing the cells to remain adhered to each other.

Pictures F–F' '(as well as in Figure 4G–L): There are discrepancies between pictures and quantifications (a 6 fold increase in the PSC>rel RNAi context compared to the control is not observed in the pictures).

We want to draw the Reviewer's attention towards the fact that the imaging has been done on live tissues to preserve and document the dynamic and fragile filopodial extensions. As soon as we put the coverslip in live conditions, the pressure created tends to detach the niche cells from each other in the control lymph glands. However, the niche cells upon Relish loss remain attached to each other. This observation can be explained by our results, which demonstrate that loss of Rel from the niche cells increases cortical actin accumulation, forcing the cells to remain adhered to each other.

Figure 5 H–L: the control, corresponding to PSC> bskDN (pictures and quantifications) is missing. Figure F5 O–Q, the control, tak1/+ (pictures and quantifications) is missing.

In the previous version, we had not included these pictures due to scarcity of space in the respective panels. In the current version, we have added few more supplementary figures to accommodate the figures into main panel. In the current manuscript, PSC>bskDN and *tak1*/+ images and quantitation has been included in Figure 5 (Figure 5F-F' and H for bskDN and Figure 5U and W for *tak1*/+).

Figure 5 L–L', should be replaced (blurry picture and migrating PSC cells).

This is a rescue experiment where the downregulation of excess cortical actin accumulation has enabled normal niche cell behavior. Kindly refer to the previous response towards migrating PSC.

What about MZ and crystal cell differentiation in these different conditions ( i.e: UAS–bskDN, UAS bskDN, Rel RNAi; tak1/+ and tak1 , rel RNAi)?

We have included the status of MZ in these two genotypes in the Figure 5 figure supplement 2. As evident from the figures, *tak1* (classical loss of function) and *JNK* loss from the niche has a subtle effect on the progenitor number assayed by Shg.

Since Rel loss from the niche does not affect crystal cell index, this aspect has no direct relationship with the data presented. However, we made an effort to address this keeping your suggestion in mind. The results related to crystal cell index are presented below. Both *tak1* (classical loss of function, Author response image 7) and bsk loss (Author response image 8) from the niche slightly decreases the crystal cell number.

**Author response image 7. sa2fig7:** Slight decrease in Crystal cell index was observed in tak12 (C) and tak12; Rel RNAi (D) compared to control (A) whereas no significant change was observed in Rel RNAi (B). E. Statistical analysis of the data from A-D (n=10 P-value=1.5 x10-2 for control versus tak12 and P-value=7.4x10-2 control versus tak12; Rel RNAi, two tailed students t-test).

**Author response image 8. sa2fig8:** Slight decrease in Crystal cell index was observed in bskDN (C) and bskDN; Rel RNAi (D) compared to control (A) whereas no significant change was observed in Rel RNAi (B). E. Statistical analysis of the data from A-D (n=10 P-value=8.5 x10-2 for control versus bskDN and P-value=6.5x10-2 for control versus bskDN; Rel RNAi, two tailed students t-test).

–Epistasis between the JNK pathway and Hh, and the JNK pathway and wg must be analysed. This is essential to support the models given in Figure 7.

We have done the epistatic analysis for Wingless and *JNK*. We have down-regulated wg function in UAS-*hep* genetic background and found a rescue in niche proliferation (Figure 5 figure supplement 1J-N).

Our earlier genetic analysis has established that *JNK* directly affects actin accumulation in the niche (UAShep^act^ and Ectopic activation of *JNK* upon Rel loss from the niche). The outcome of upregulated actin leads to trapping of Hh within the niche, thereby causing progenitors to undergo precocious differentiation.

Figure 6: The demonstration that Ecdysone controls relish transcription in the PSC is lacking Analysis of Relish protein levels is not enough, since post transitional regulation has been described for Relish. relish transcription must be analyzed in PSC>EcRDN LGs. Furthermore, epistasis experiments are required to firmly establish a functional link between Relish and ecdysone.

We have done rescue experiments in which we over-expressed Relish in EcR loss genetic background and found a rescue in niche proliferation (Figure 6 Q-U) and differentiation defects (Figure 6 figure supplement 2A-E) associated with EcR loss. Further, we also did the whole mount IF along with FISH and found that compared to control; there is a reduction in Rel transcripts level when EcR is downregulated from the niche (Figure 6 figure supplement 2K-L'').

Thanks for your input. With newly included epistatic analysis and dual antibody-FISH results, have strengthened our manuscript to a large extent.

Figure 7: The link between Relish in the PSC and its role in LG hematopoiesis in response to bacterial infection is not yet clearly assessed. That an increase in blood cell differentiation occurs in the LG in response to bacterial infection has already been described (Khadilkar et al;, 2017). This reference should be cited in the manuscript. Whether the bacteria–induced decrease of Relish in the PSC is involved in the LG stress hematopoiesis is not established here since there are many discrepancies between LG phenotypes in PSC>rel RNAi larvae and in response to bacterial infection and rescue experiments are lacking. While in both cases there is an increase in plasmatocyte differentiation, there is no difference in PSC size in response to bacterial infection (which differs from PSC>rel RNAi LGs), and there is no LG disruption in PSC>rel RNAi larvae (which differs from bacterial infection). Furthermore, the LG disruption in response to bacterial infection is a novel data, but unfortunately it is not properly documented in this manuscript. Finally, in bacterial infection experiments, no data relative to a role of Hh or JNK pathway in PSC cells have ever been described so far.

Thanks for your inputs. We have included the references in the main text of the current version.

However, we disagree with the statement that there are discrepancies in the phenotype of Rel loss data compared to infection. Kindly note the following reasons:

1. Similar to loss of Rel from the Niche, *JNK* signaling is evoked during infection (please refer to Figure 7 figure supplement 1A-B').

2. In context to niche cell number, the lymph glands in the infection experiments were analyzed 4 hours post insult, a short time window to expect a similar degree of proliferation compared to that of Rel knockdown from the niche. However, even in this short window of insult, we encounter a significant and consistent increase in the niche cell numbers in infected samples compared to mock (Figure 7 figure supplement 1C-E).

3. In infected samples, we do see lymph gland disruption beyond 4 hours. Upon loss of Rel from the niche, we see a similar response in few cases where there is extreme differentiation accompanied by peeling off the LG around 96 AEH (Please see Author response image 9). Since most LGs were intact at 96AEH, we did not include the LG with peeled off phenotype as the representative image in the current manuscript. However, we are open to suggestions from your end and, if required, will include this in the current version.

**Author response image 9. sa2fig9:** A-D. Compared to control (A) ectopic differentiation and peeling off of lymph glands was observed in infected (B) as well as Rel loss samples (C).

Is the transcription of relish or the stability of the protein that is affected in response to bacterial infection? In Figure 7 D: >UAS –Rel 68KD, without infection, is Relish protein detected by immunostaining in PSC cells? This would clarify whether Relish protein stability is dependent on bacterial infection.

Thank you for your suggestion. Antp-GFP >UAS-Rel68KD was capable of enriching Rel expression in the niche (See results in Author response image 10). However, on infecting the same genotype, we encountered a drop in Rel expression Please refer to Figure 7H-H' of the current manuscript. Thus, we can infer that forced expression of Rel in the niche was not capable of withstanding the immune challenge. These results indicate that the stability of the protein is at stake during infection

**Author response image 10. sa2fig10:** A-B: Rel expression in the niche was observed in control as well as in UAS-Rel68KD tissues.

Figure 7 B–D: Both quantifications of PSC cell numbers and Relish levels are missing.

Thanks for your suggestions. We have included these missing information’s in the current version.

The niche cell proliferation data (Figure 7 figure supplement 1C-E) and Relish levels (Figure 7A-D) has been included in the modified manuscript.

Figure 7 H–J': Plasmatocyte quantification is missing. Analyses and quantifications of crystal cells and MZ progenitors must also be provided.

We have included the quantitation of progenitors and plasmatocytes in the current version. Compared to control, there was significant decrease in DE-Cad (Shg) positive progenitor cells (Figure 7I-K) and a concomitant increase in the plasmatocyte differentiation (Figure 7L-N) was observed in infected lymph gland compared to mock. The crystal cell index is provided in Author response image 11. As previously described (Khadilkar et al., 2017), we also see an increase in their number upon infection.

**Author response image 11. sa2fig11:** Significant increase in crystal cell index was observed in infected samples compared to sham (n=9, P-value =1. 8x10-3, two tailed students t-test).

Models in Figure A and K are not correct, since many epistasis experiments are missing.Furthermore, an arrow between Wg in the PSC and MZ progenitors is missing, since it has been previously established that the Wg pathway in the PSC controls LG hemocyte differentiation (please see Sinenko et al., 2009).

We have enriched our findings with multiple epistatic analyses and did the necessary modifications in the current version.

Figure 8 : Are there any functional links between filopodia, the actin skeleton and Hh signaling in the LG ? This point should be further addressed.

As mentioned in the manuscript, there are multiple reports regarding the role played by cellular cytoneme like filopodia in Hh delivery from the niche to progenitor cells. In addition to our data, works from the laboratory of Prof Schulz (Tokusumi et al., 2012, Tokusumi et al., 2010) have demonstrated the functional link between filopodia and Hh delivery in the LG.

For the PSC>rel RNAi context, both in the summary and in point e in the legend Figure 8, we read "precocious differentiation of the blood progenitors". This point should be better established. Indeed, systematic analyses of MZ markers in the different genetic contexts are required (see comments above).

We have done Shg immunostaining and found a significant decrease in the number of progenitor cells compared to the control (Figure 1H-J). Additionally, integrin β-PS, another progenitor marker expression was significantly lower (Figure 1 figure supplement 1I-J') in loss of Rel loss from the niche compared to control.

[Editors’ note: what follows is the authors’ response to the second round of review.]

Essential revisions:General comments by the editor following the discussion with the reviewersBoth reviewers feel that this paper warrants publication in eLife.However , they think that additional new data are needed to fully support the conclusion on the role of PSC fillopodia in signaling. At the very least, the authors must establish whether there is a functional relationship between Relish and actin binding protein levels in the PSC. Rescue experiments of relish knock down, by decreasing "actin protein" and looking at lymph gland homeostasis, fillopodia formation and Hh diffusion, have to be provided.No functional relationship has been established for the role of Relish in the PSC in response to bacterial infection Thus, the authors should be more cautious, and both in the manuscript and the summary the text relative to this point has to be modulated. Furthermore, the authors should discuss the discrepancy between their model, where in response to bacterial infection Hh is trapped within the niche and unable to diffuse, versus the model proposed by (Khadilkar et al., 2017) where bacterial infection causes the breakdown of the permeability barrier in PSC cells.To minimize the second round of revision, the reviewers suggested that the authors could also tone down the statements on the filopodia–Hh interaction and the role of Rel in the bacterial infection while adding possible additional works.Essential revisions:1) As I already suggested in the previous comment, the authors should impair PSC fillopodia formation by modifying actin binding proteins (F actin, singed,dia,…) and analysing both lymph gland defects and Hh diffusion.

We have downregulated *dia* from the PSC/niche, and the lymph gland defects observed has been included in the revised version Figure 4 figure supplement 1. Since the filopodial length and number were significantly compromised (A-D), we observed a defect in the transport of extracellular Hh (E-G). As a consequence we see an increased differentiation (J-K and M) and decline in progenitors index (H-I in L). We have also included this in the main text of the revised version. Please refer to page 12 and Line 276-284.

It is also necessary to establish whether there is a functional relationship between Relish and actin binding protein levels in the PSC. Rescue experiments of relish knock down, by decreasing "actin protein" and looking at lymph gland defect, fillopodia formation and Hh diffusion are missing.

Thanks for this suggestion. Since a robust upregulation of “actin binding protein” Ena is observed upon Rel loss from the PSC, we decided to down-regulate Ena in conjunction with Rel loss from PSC to address their relationship.

Loss of both Rel and Ena from the niche led to a partial rescue in the defects related to filopodia, Hh dispersal, and differentiation, which are otherwise observed upon Rel loss. Just by modulating one “actin binding protein”, getting a partial rescue indicates a functional relationship between Rel and actin-binding proteins.

Please refer to Figure4 figure supplement 3 and Page: 13 and Lines: 296-300 in the main text of the revised manuscript.

In Figure 4 Sup1: In PSC>UAS Dia RNAi, what about fillopodia formation and Hh diffusion ? Does decreasing dia in PSC cells in relish knocked down condition restore fillopodia formation, Hh diffusion and lymph gland cell homeostasis? This is an important issue than remains to be addressed.

As mentioned above we have performed this set of experiments. Please refer to the revised Figure 4 Figure supplement 1. As expected loss of Dia from PSC affects filopodia formation. As a result both defects in Hh transport and progenitor maintenance is evident upon *dia* loss. Based on the result we can speculate that loss of *dia* along with *Rel* would enhance the phenotype.

Figure 4A–C: when relish is knocked down in the PSC, is Hh the increase in PSC cells due to a defect of Hh diffusion and/or a decrease in Hh transcription in those cells? This has to be clarified.

We have now looked at the transcription readout of Hh: *hhF4f*GFP expression, which seems to be also elevated in niches where Rel is downregulated. Please refer Author response image 12:

**Author response image 12. sa2fig12:** A-B'. Loss of Relish function from the niche resulted in upregulation of *hh* transcription (A- A') (marked by *hh*-F4f GFP) compared to control niches (B-B')C. Statistical analysis of the data provided in A-B' (n=15, P-value= 2.2x10-11, two tailed Students t-test).

2) The role of Relish in the PSC in response to bacterial infection is based on correlation, since rescue experiments are missing and thus no functional relationship has been established yet.

In order to perform a rescue experiment, a bacterial infection was performed on individuals where *UAS-Rel68kD* was overexpressed in the niche. However, much to our surprise, the upregulated Rel gets degraded at the protein level during bacterial infection. This itself unraveled a niche specific-regulation on Rel during infection. Therefore, the conventional rescue experiment is not possible. Though the candidate that breaks the maintenance circuit remains to be identified, nonetheless, our study illustrates that the hematopoietic niche can sense the physiological state of an animal to facilitate a transition from normal to emergency hematopoiesis via Rel.

Furthermore, whereas bacterial infection leads to the increase in crystal cell differentiation (Khadilkar et al., 2017), relish knock down in PSC cells has no impact on crystal cell number. Thus, the authors should be more cautious, and both in the manuscript and the summary the text relative to this point has to be modulated.

Bacterial infection is a systemic challenge that can have multiple impacts on the lymph gland. One such impact is what we have shown through our study where a septic injury resulted in the loss of Relish from the hematopoietic niche. Hence, knocking down the function of just one gene (in our case Relish) cannot mimic all the phenotypes associated with the systemic immune challenge.

Moreover, we also speculate that in Relish loss from the niche, there can be loss of multiple positive as well as negative regulators involved in crystal cell differentiation, thereby dampening individual outputs.

Finally, the authors should discuss the discrepancy between their model where in response to bacterial infection Hh is trapped within the niche and unable to diffuse, versus the model proposed by (Khadilkar et al., 2017) where the bacterial infection causes the breakdown of the permeability barrier in PSC cells.

We want to draw the reviewer's attention towards the fact that comparison with the mentioned paper is not possible. This is because Khadilkar et al. has not assayed the status of Hh in their case; instead, their case emulates a niche that has decanted signals that can be both of maintenance and differentiation.

While in our case, cytoskeletal rearrangements in the niche prevent the dispersion of Hh or the dispersion of signals from the niche. We show that the trapped Hh is not sensed by the progenitors, which fails to maintain themselves and thus differentiates.

In the cited paper, seeping off too many differentiation signals (Please refer to the Discussion section of Khadilkar et al.) leads to ectopic differentiation.

If maintenance and differentiation are both sides of a coin: Khadlikar et al., describes scenario elevated differentiation signaling that pushes progenitor towards differentiation. On the other hand, we illustrate how the failure of maintenance can also be a reason for differentiation. Moreover, at this time point, we want to draw your attention to that signals for differentiation are not only elicited from Niche, the progenitors also generate them. As advised by the reviewer, we have discussed Khadilkar et al. findings in the Discussion section of the revised manuscript.

Please refer to page: 22 and Lines: 530-537.